# Solving Normalized Cut Problem with Constrained Action Space

## Abstract

We address the problem of Normalized Cut (NC) in weighted graphs where the shape of the partitions follow an apriori pattern, namely they must approximately be shaped like rings and wedges on a planar graph. Classical methods like spectral clustering and METIS do not have a provision to specify such constraints and neither do newer methods that combine GNNs and Reinforcement Learning as they are based on initialization from classical methods. The key insight that underpins our approach, Wedge and Ring Transformers (WRT), is based on representing a graph using polar coordinates and then using a multi-head transformer with a PPO objective to optimize the non-differential NC objective. To the best of our knowledge, WRT is the first method to explicitly constrain the shape of NC and opens up possibility of providing a principled approach for fine-grained shape-controlled generation of graph partitions. On the theoretical front we provide new Cheeger inequalities that connect the spectral properties of a graph with algebraic properties that capture the shape of the partitions. Comparisons with adaptations of strong baselines attest to the strength of WRT.

## 1 Introduction

Reinforcement Learning (RL) has emerged as a powerful heuristic for tackling complex combinatorial optimization (CO) problems Grinsztajn (2023); Wang & Tang (2021); Mazyavkina et al. (2021). Two key insights underpin the use of RL in CO: first, the search space of CO can be encoded into a vector embedding; second, gradients can be computed even when the objective is a black-box function or non-differentiable. A significant advantage of RL frameworks is that once trained, they can solve new instances of CO problems without starting from scratch Dong et al. (2020).

In this work we present another dimension of the use of transformed-based RL for graph partitioning, namely the ability to encode and optimize complex partition shapes that are part of the problem specification. We focus on the Normalized Cut (NC) of a graph, which is suitable to balance the simulating traffic on road networks. While our use case is inspired by a specific problem in road vehicle traffic simulation, our approach is general and can be applied in many other scenarios where shapes of graph partitions are application dependent.

**Motivational Use Case:** Road networks in modern cities are often organized as concentric rings of roads centered at a city downtown followed by wedge structures connecting the outer ring. For microscopic traffic simulation, where the movement of every vehicle is modeled in a simulator, it often becomes necessary to partition the road network and assign each partition to a separate simulator in order to reduce the overall simulation time. We thus want to ensure that the partitions apriori respect the natural physical topology of the road network. Directly using classical approaches like METIS, spectral clustering or modern GNN based RL solutions provide no provision to constrain the generation of partition shapes justifying the need for a new approach.

**Ring and Wedge Representation:** The key insight of our is to convert complex graph structures into simpler representations (either as a line or a circle), reducing the complexity of the partitioning problem. This transformation makes the graph more amenable to being processed by Transformer-based models, which excel at sequential data processing. In the ring transformation, nodes are projected onto the $x$-axis according to their radial distance from the center, preserving the node order and partitioning properties. Similarly, in the wedge transformation, nodes are projected onto a unit

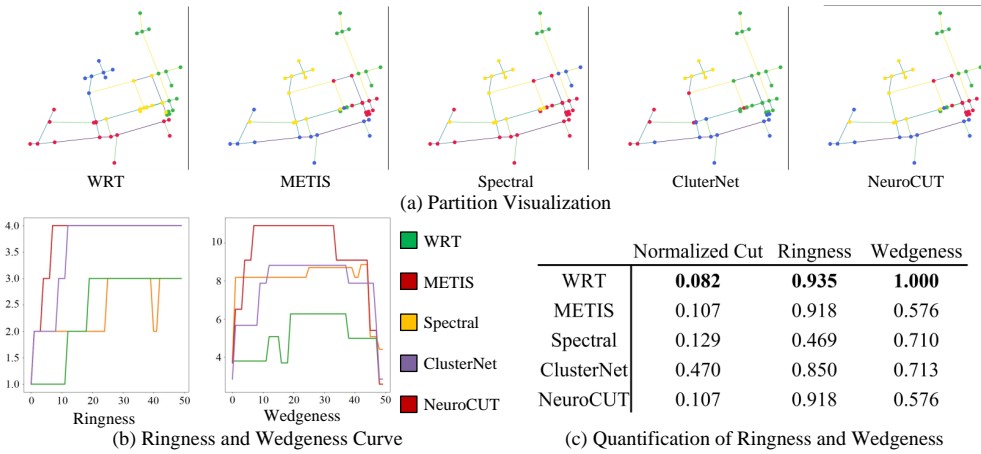

Figure 1: Compared with other methods, WRT has the minimal Normalized Cut, and also achieves the highest Ringness and Wedgeness (which is formally defined in Section 3). NeuroCUT is initialized by METIS partition, and fails to find a better one, which causes the same result.

circle, focusing on their angular positions. These transformations allow us to apply Transformers, which can scale more effectively to large graphs compared to traditional GNNs.

After transforming the graph, we apply Proximal Policy Optimization (PPO)Schulman et al. (2017) to solve the partitioning problem. Our approach leverages the ability of Transformers to capture both local interactions and global patterns across the entire graph. We demonstrate that our method outperforms existing RL-based and traditional methods, particularly in handling weighted planar graphs. In additional to optimizing Normalized Cut, we explicitly measure the *ringness* and *wedgeness* of the generated partitions. We give performance visualization in Figure 1. In Figure 1(a), a snapshot of the partitions generated by different methods shows that other methods except our proposed method WRT tend to mix nodes from different partitions, resulting in high Normalized Cut. Figures 1(b) and 1(c) introduce Ringness and Wedgeness metrics to evaluate how closely a partition aligns with ring and wedge structures. Our proposed method, WRT, achieves the lowest Normalized Cut while maintaining the highest Ringness and Wedgeness scores.

Our main contributions are as follows:

- A novel RL-based approach to minimize Normalized Cut on planar weighted graphs.
- The introduction of the ring-wedge partitioning scheme (WRT), which simplifies graph structures for more efficient processing by Transformer models, and use two-stage training process which improves partitioning performance and stability.
- Our extensive experiments on synthetic and real-world graphs show that our algorithm have the best performance and scales to graphs with different sizes effectively.

## 2    RELATED WORK AND PRELIMINARIES

### 2.1    GRAPH PARTITIONING

Graph partitioning Buluç et al. (2016) is widely used in graph-related applications, especially for enabling parallel or distributed graph processing. Partitioning a graph into $k$ blocks of equal size while minimizing cuts is NP-complete Hyafil & Rivest (1973). Exact methods focus on bipartitioning Hager et al. (2009) or few partitions ($k \leq 4$) Ferreira et al. (1998), while approximate algorithms include spectral partitioning Donath & Hoffman (1973) and graph-growing techniques George & Liu (1981). More powerful methods involve iterative refinement, such as node-swapping for bipartitioning Kernighan & Lin (1970), extendable to $k$-way local search Karypis & Kumar (1996).

Other approaches include the bubble framework Diekmann et al. (2000) and diffusion-based methods Meyerhenke et al. (2009); Pellegrini (2007). State-of-the-art techniques rely on multilevel partitioning Karypis & Kumar (1999), which coarsen the graph and refine the partition iteratively.

The most well-known tool is METIS Met (2023); Karypis & Kumar (1999), which uses multilevel recursive bisection and $k$-way algorithms, with parallel support via ParMetis Par (2023). Other tools include Scotch sco (2023); Pellegrini (2007) and KaHIP Sanders & Schulz (2011) use various advanced techniques. However, these methods are suboptimal for minimizing normalized cuts in spider-web-shaped structures common in urban traffic planning.

## 2.2 ML-BASED GRAPH PARTITIONING ALGORITHMS

Recent research has explored machine learning methods for graph partitioning, particularly using GNNs. GNNs aggregate node and edge features via message passing. In Gatti et al. (2022a), a spectral method is proposed where one GNN approximates eigenvectors of the graph Laplacian, which are then used by another GNN for partitioning. The RL-based method in Gatti et al. (2022b) refines partitions in a multilevel scheme. NeuroCUT Shah et al. (2024) introduces a reinforcement learning framework that generalizes across various partitioning objectives using GNNs. It demonstrates flexibility for different objectives and unseen partition numbers. ClusterNet Wilder et al. (2019) integrates graph learning and optimization with a differentiable k-means clustering layer, simplifying optimization tasks like community detection and facility location. However, neither of these methods handles weighted graphs, making them unsuitable in our scenarios.

Although GNNs excel at aggregating multi-hop neighbor features, they struggle to globally aggregate features without information loss, which is critical for combinatorial problems like graph partitioning. Our work addresses these limitations by introducing graph transformation methods and applying Transformer to learn global features.

## 2.3 REINFORCEMENT LEARNING

In our work, we use Reinforcement Learning, specifically PPO to train the model with non-differential optimizing targets. Proximal Policy Optimization (PPO) Schulman et al. (2017) is a widely-used RL algorithm that optimizes the policy by minimizing a clipped surrogate objective, ensuring limited deviation from the old policy $\pi_{\text{old}}$. The PPO objective maximizes $\mathbb{E}_t \left[ \min(r_t(\theta)A_t, \text{clip}(r_t(\theta), 1 - \epsilon, 1 + \epsilon)A_t) \right]$, where $r_t(\theta) = \frac{\pi_\theta(a_t|s_t)}{\pi_{\theta_{\text{old}}}(a_t|s_t)}$ and $A_t$ is the advantage.

## 3 PROBLEM DESCRIPTION

Let $G = (V, E, W, o)$ be a weighted planar graph, with vertex set $V$, edge set $E$, edge weights $W$, and a predefined center $o$. A $k$-way partition $P$ of $G$ is defined as a partition $\{p_1, ..., p_k\}$ of $V$, where $\bigcup_{i=1}^{k} = V$ and $\forall i \neq j, p_i \cap p_j = \varnothing$.

We introduce the definition of the **Normalized Cut** as follows: For each partition $p_i$, we define

$$Cut(G, p_i) = \sum_{u \in p_i \otimes v \in p_i} W(e_{u,v}) \quad Volume(G, p_i) = \sum_{u,v \in p_i} W(e_{u,v}) + Cut(G, p_i), \quad (1)$$

where $\otimes$ represents the XOR operator. The *normalized cut* of a partition $P$ on graph $G$ is then defined as

$$NC(G, P) = \max_{i \in \{1..k\}} \frac{Cut(G, p_i)}{Volume(G, p_i)}. \quad (2)$$

We aim to find partitions that minimize the normalized cut, a known NP-complete problem, and thus we focus on approximate solutions. The goal is to learn a mapping function $f_\theta(G) = P$ that minimizes $NC(G, P)$.

Instead of considering the entire space of possible partitions, we restrict our attention to partitions with specific structures, namely those where each partition is either ring-shaped or wedge-shaped. We also allow for "fuzzy" rings and wedges, where a small number of nodes are swapped to adjacent

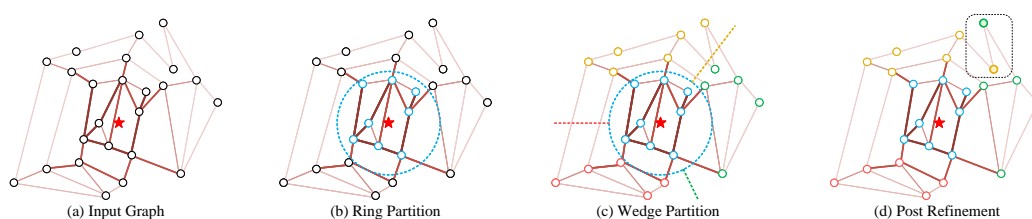

(a) Input Graph      (b) Ring Partition      (c) Wedge Partition      (d) Post Refinement

Figure 2: Graph partitioning with Ring and Wedge to minimize the Normalized Cut. We firstly do ring partitions as (b), to choose different radii to partition the graph into rings. Then for the out-most ring, we do partitions based on different angles as (c). Finally, we do post refinement to improve the final partition performance as (d).

partitions. This relaxation helps achieve partitions with a smaller normalized cut, particularly for graphs derived from real-world applications.

Our partitioning strategy follows a three-step process: first, we perform a ring partition on the entire graph, then we apply a wedge partition to the outermost rings, and finally, we refine the resulting partitions to further reduce the normalized cut. Figure 2 illustrates these three steps.

**Ring Partition:** A Ring Partition of the graph $G$ with respect to the center $o$, denoted by $P^r$, divides $G$ into $k_r$ distinct concentric rings. Define the radii as $0 = r_0 \leq r_1 \leq r_2 \leq \cdots \leq r_{k_r-1} < r_{k_r}$. These radii partition $G$ into $k_r$ rings, where the $i$-th ring, denoted as $p_i^r$, contains all nodes with a distance to the center $o$ between $r_{i-1}$ and $r_i$.

**Wedge Partition:** A Wedge Partition, denoted as $P^w$, divides the outermost ring $p_{k_r}^r$ into multiple wedge-shaped sections. The partitioning angles are given by $0 \leq a_1 \leq a_2 \leq \cdots \leq a_{k_w} < 2\pi$. These angles split $p_{k_r}^r$ into $k_w$ wedge parts, where the $i$-th wedge, $p_i^w$, contains the nodes whose polar angles are between $[a_i, a_{i+1})$, except for the wedge $p_{k_w}^w$, which contains nodes whose angles fall within either $[0, a_1)$ or $[a_{k_w}, 2\pi)$.

This type of partition divdes the graph into $k_r - 1$ inner rings and $k_w$ wedges on the outermost ring (see Figure 2). Specifically, if $k_r = 1$, the entire graph is partitioned solely by wedges and, conversely, if $k_w = 1$ the graph is partitioned solely by rings. For simplicity, when a graph $G$ is partitioned by a Ring-Wedge Partition with $k_r$ and $k_w$, we define $k = k_r + k_w - 1$, with $p_k = p_k^r$ when $k < k_r$, and $p_k = p_{k-k_r+1}^w$ when $k >= k_r$. And we define the total partition strategy as $P = \{p_1, ..., p_k\}$.

We also propose the Ringness and Wedgeness to evaluate whether a partition is close to the ring shape or wedge shape. The definition of Ringness and Wedgeness can be found in the Appendix.

Besides the practical aspects, partitions structured as a combination of ring and wedge subsets seem also theoretically well behaved. For example, on a simple class of graphs, they satisfy bounds similar to the ones that are satisfied by partitions achieving minimum normalized cut. In the next section, we provide these bounds for the class of spider web graphs.

## 4 CHEEGER BOUND FOR RING AND WEDGE PARTITION

In the graph partitioning context there exists bounds on the Cheeger constant in terms of the normalized Laplacian eigenvalues, see for example Chung (1997) for bisection and Lee et al. (2014) for more general k-partitions. Intuitively, the Cheeger constant measures the size of the minimal "bottleneck" of a graph and it is related to the optimal partition. Since we consider a subset of all the possible partition classes, namely ring and wedge, we show that the normalized cut defined in equation 2 satisfies bounds similar to the classical case in the case of unweighted spider web graphs. Despite being a simpler class of graphs, these bounds give a theoretical justification of the normalized cut definition equation 2 and the ring-wedge shaped partition. (see the proof in Appendix).

**Definition:** *Let $G_{n,r}$ be an unweighted spider web graph with $r$ rings and $n$ points in each ring,*

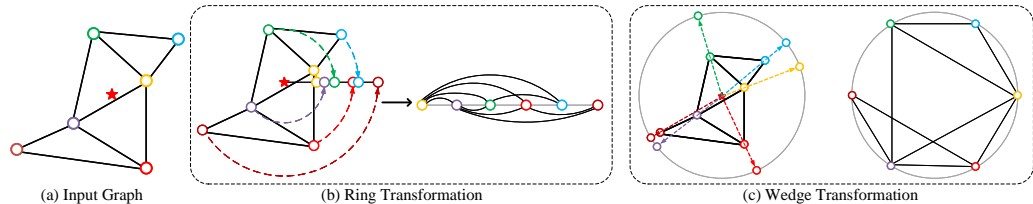

(a) Input Graph                (b) Ring Transformation                (c) Wedge Transformation

Figure 3: Example of Wedge Transform and Ring Transform. In Wedge Transform, nodes are projected to a circle, then the difference of angles of adjacent nodes are adjusted to the same. In Ring Transform, nodes are projected to a line. The edge connections and their weights are not changed in both transformation.

*and $k$ be an integer. Define the wedge and ring Cheeger constants as:*

$$\phi_{n,r}(k) = \min_{\substack{P=V_1 \cup \cdots \cup V_k \\ \text{wedge partition}}} NC(G_{n,r}, P) \qquad \psi_{n,r}(k) = \min_{\substack{P=V_1 \cup \cdots \cup V_k \\ \text{ring partition}}} NC(G_{n,r}, P). \qquad (3)$$

**Proposition 1** *Let $G_{n,r}$ be a spider web graph with $r$ rings and $N$ nodes in each ring. Let $\lambda_k^C$ and $\lambda_k^P$ be the eignevalues of the circle and path graphs with $n$ and $r$ vertices respectively. Then*

$$\phi_{n,r}(k) \leq \frac{2r}{2r-1}\sqrt{2\lambda_k^C}, \quad 2 \leq k \leq n \qquad \psi_{n,r}(k) \leq \sqrt{2\lambda_k^P}, \quad 2 \leq k \leq r. \qquad (4)$$

## 5 METHODOLOGY

To elaborate on our approach, we begin by introducing the reinforcement learning environment settings, then we provide a general overview of the agent's role and its interaction with the environment to achieve the final partition. We then dive into the detailed structure of the method. Finally, we discuss training methodologies and post refinement methods aimed at enhancing performance.

For simplicity, we will pre-define the ring partition number $k_r$ and wedge partition number $k_w$. When $k$-partitioning a graph, we will enumerate all possible ring partition numbers, then select the one with minimum normalized cut as the result.

### 5.1 REINFORCEMENT LEARNING ENVIRONMENT

We primarily employ reinforcement learning methods to address the ring-wedge partitioning problem. The observation space, action space and reward function are defined in the following. The agent's final goal is to maximize the reward through interactions with the environment described above.

**Observation Space** The observation space $S$ contains the full graph $G$, the expected ring number $k_r$, wedge number $k_w$, and the current partition $P$, denoted by $S = \{G, k_r, k_w, P\}$.

**Action Space** The agent needs to decide the next partition as action. If it is a Ring Partition, the action is the radius of next ring, if it is a Wedge Partition the action is the partition angle of the wedge.

$$A = \left\{ \begin{array}{ll} r & \text{if currently expects a ring partition} \\ a & \text{if currently expects a wedge partition} \end{array} \right.$$

**Reward Function** When the partition is not over, we use 0 as reward. When the partition is over, i.e. current partition number achieves pre-defined total partition number, we calculate the Normalized Cut, and use the negative of it as the reward, as we need to minimize the Normalized Cut, i.e., $r = -NC(G, P)$.

### 5.2 GRAPH TRANSFORMATION

In previous deep learning based graph partitioning methods, most of them chose the combination of GNN and Reinforcement Learning. However, GNN suffers from only being able to aggregate global structure of the graph, hence they need an initial partition and do fine-tuning on it, which is not capable in our situation, as we want the model give ring and wedge partition results directly.

Recently, Transformer achieves great success in various areas, it uses Multi-Head Attention to exchange information globally, and shows superior performance in various tasks. In our problem, we need the model to learn the global view of the graph, and we naturally choose Transformer as the base structure. However, Transformer typically takes sequential input, which is not capable for graphs. Instead of directly encode graph nodes to Transformer, we apply two transformations, Ring Transformation and Wedge Transformation, to the graph. The new graphs are equivalent with original graph when performing Ring Partition or Wedge Partition, but is re-organized into a sequential representation, and is able to input to Transformer.

### 5.2.1 RING TRANSFORMATION

Since the ring partition should not change when rotating the graph around the center $o$, we can project each node onto the $x$-axis. More precisely, if a node has polar coordinates $(r, X)$, the projection will map it onto the node with coordinate $(r, 0)$. Note that this transformation does not change the order of the nodes or the partitions. Figure 3 (b) illustrates the projection onto the line. Then we can find that when the order of nodes on the line are not changed, we can adjust the radius of any point, and the partition results on new graph are the same as old ones. When we apply the conclusion above, we can transform a normal graph into a simplified one, that every nodes are with coordinate $(X, 0)$, where X is the radius order of the node along all nodes. The transformation results is shown in the right of Figure 2 (b).

### 5.2.2 WEDGE TRANSFORMATION

Similar to Ring Transformation, we find that when doing wedge partition, the node radius has no effect, and only the node angle is considered. We project all nodes into a unit circle which has $o$ as its center. Hence, if $(r, X)$ are the polar coordinates of a node, its projection will have coordinates $(1, X)$. After projection, we can also change the angles of nodes. If the angle order of a node is $X$ from $N$ nodes, its new position is on $(1, \frac{2\pi X}{N})$ with polar coordination. The Transformation process is illustrated in Figure 3 (c).

After transformation, nodes of the graph lie on a line or on a circle, hence we can treat the graph as a sequential input. We can also find that for actions that split nodes $i$ and $i + 1$ into two partitions will perform exactly the same final partition results. As the result, we can convert the continuous action space into discrete ones to decrease the learning difficulties. New action $A_i$ means split node $i$ and $i + 1$ into two partitions.

### 5.3 RING WEDGE PARTITION PIPELINE

The graph partition pipeline of the Wedge Ring Transformer (WRT) is illustrated in Figure 4 (a). It sequentially determines partitions through Ring and Wedge Transformations, predicting the next ring radius or wedge angle until the target partition count is achieved. The model consists of two components for ring and wedge partitions with similar structures but distinct weights.

**Transformation:** The appropriate transformation (ring or wedge) is applied based on current requirements.

**Pre-Calculation:** Essential computations on the transformed graph include: (1) Cut Weight $C_i$: Sum of edge weights crossing between nodes $i$ and $i + 1$. (2) Volume Matrix $V_{i,j}$: Total weight of edges covered between nodes $i$ and $j$ (where $i < j$).

**Wedge Ring Transformer:** The Transformer processes node embeddings from the pre-calculation phase and the current partition status, as depicted in Figure 4 (b).

**PPO Header:** After receiving node embeddings, the PPO header extracts action probabilities and critic values. The actor projection header maps hidden size $h$ to dimension 1, followed by a Softmax layer for action probabilities. Value prediction uses Self-Attention average pooling on node embeddings and projects from $h$ to 1. The PPO is employed to execute actions recursively until the graph is fully partitioned.

During the Ring Partition phase, a dynamic programming algorithm calculates the optimal partition when the maximum radius and total ring count are fixed, with a complexity of $O(n^2 k)$. Thus, the WRT determines the maximum radius for all ring partitions only once. The pseudo-code is available in the Appendix.

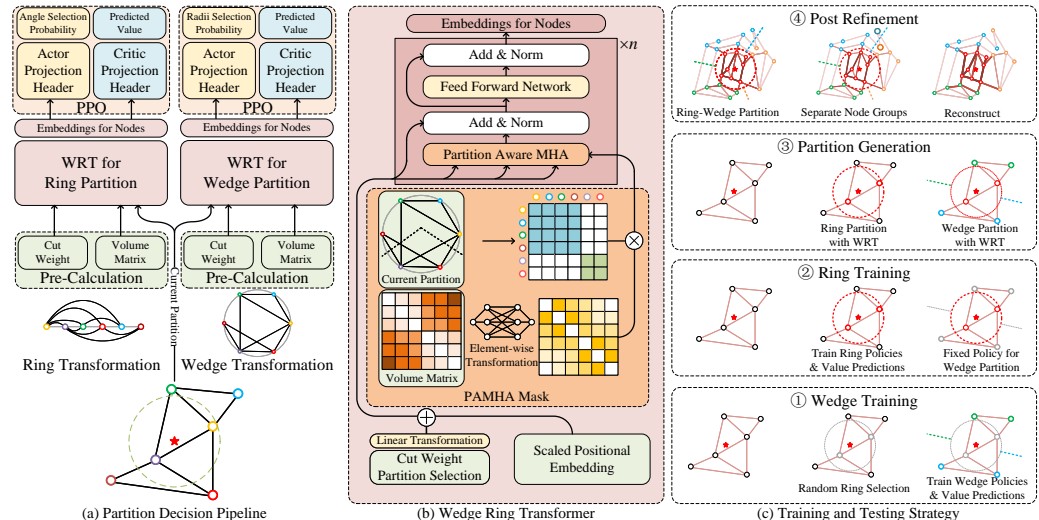

Figure 4: Framework and stages of the Wedge-Ring Transformer (WRT). (a) WRT first applies Ring and Wedge Transformations, followed by pre-calculation to obtain cut weights and the volume matrix. The processed data generates node embeddings for action probabilities and predicted values via actor and critic projection headers. Modules for ring and wedge partition share structures but differ in weights. (b) Detailed structure of WRT, using cut weights with positional embeddings as input, followed by transformer layers. Volume matrix and position information serve as attention masks in the MHA layer, ensuring focus on nodes within the current partition. (c) WRT pipeline from training to testing. Initially, the wedge partition strategy is trained with a random approach for the ring partition. The wedge part is fixed while training the ring part, excluding its critic projection header. During testing, the WRT sequentially determines ring radius and wedge angle, refining the final partition using a post-refinement algorithm.

## 5.4 WEDGE RING TRANSFORMER (WRT)

WRT utilizes a Transformer backbone to leverage information from transformed graphs, enabling it to handle varying node counts and enhancing its scalability for diverse applications without the need for fine-tuning after training. The Transformer architecture is illustrated in Figure 4 (b).

WRT processes inputs from the Pre-Calculation module, specifically Cut Weight and Volume Matrix, along with the Current Partition from the input graph. These are fed into $n$ Transformer blocks, yielding node embeddings from the final hidden state. To effectively manage Current Partition, we represent each node's selection status Partition Selection with a 0-1 array, then it is combined with Cut Weight and transformed through a linear layer to generate hidden states, which are subsequently augmented with positional embeddings.

We introduce Partition Aware Multi-Head Attention (PAMHA) to replace the original Multi-Head Attention (MHA) layer. PAMHA incorporates the Volume Matrix and Current Partition into its attention mask. An element-wise transformation on $V$ produces an attention mask of shape $N \times N$ for PAMHA, allowing the model to learn the significance of different nodes. For Current Partition, we observe that partitions splitting between nodes $i$ and $i + 1$ do not affect the normalized cut calculations on the right of $i+1$. For instance, in the circular graph with six nodes depicted in Figure 4, partitioning between certain nodes does not alter the normalized cut of other nodes. Consequently, we create an attention mask focusing solely on the effective range of nodes. Finally, WRT outputs node embeddings, which are then input to the PPO module.

## 5.5 TRAINING AND TESTING STRATEGIES

We use a special training and testing strategies for the problem to learn better policies and give better partition results. Both training and testing contains two stages. Visualization of four stages are shown in Figure 4 (c).

### 5.5.1 TRAINING STRATEGY

With previous model design, WRT are able to dig out information effectively from a graph. However, in RL, the initial strategies are randomized, which makes it challenging to learn a good strategy, specifically ring partition and wedge partition will obstruct each other. For example, if the ring partition always selects the smallest radius as the action, the wedge partition cannot learn any valid policy because the total Normalized Cut is determined by ring partition. Training ring partition with a low quality wedge partition strategy will also face such difficulty.

To mitigate the above problem, we split the policy training into two stages, as shown in Figure 4 (c) ① and ②. In the first Wedge Training stage, we use a randomized ring selection method to replace the ring selection strategies, and only let WRT decide and train on wedge partitioning. To make the model focus on learning good wedge partition strategy, we also ignore the Normalized Cut of rings when calculating the reward. This makes the model focus on learning wedge partition strategy.

In the second Ring Training stage, we let WRT decide both ring and wedge partition. However, we find that if we allow the model to tune all its parameters, the model is likely to forget how to perform a good wedge partitioning before learning a good ring partition strategy. To avoid this, we fix the parameters of wedge partitioning modules in WRT, as WRT has learned a good wedge partition strategy with various radius. The only exception is Critic Projection Header, because in the previous stage we change it to only use the Normalized Cut of wedge partitions as the reward, which is inconsistent with current reward definition. During the Ring Training stage, two Critic Projection Headers are both re-initialized and trained. In PPO, as the strategy are only determined by actor model, allowing critic to be trainable will not affect the learned policy.

### 5.5.2 TESTING STRATEGY

After WRT is fully trained, we can directly generate partitions by WRT in Partition Generation stage, it will firstly do ring partition, then do wedge partition in sequential, as shown in Figure 4 (c) ③.

While we have proved ring and wedge partitions have the similar upper-bounds with with constraints, sometimes in real graph, ring and wedge partition may not be the optimal one as the graph has outliers when performing ring and wedge partition. We give an example in Figure 4 (c) ④, the group of two nodes are reversed when performing a pure ring and wedge partition. To mitigate such problem, we perform a Post Refinement Stage, where nodes in the same partition but not connected will be split into multiple partitions. Then we greedily choose the partition which has biggest Normalized Cut, and merge the partition into adjacent partitions. This post refinement method will decrease the outlier node number, and gives better partitions.

Finally, as the action of PPO is a policy-gradient based method, which provides an action probability distribution, and single segmentation may not yield the optimal solution directly, we can perform multiple random sampling to obtain different partitions and choose best of them.

## 6 EXPERIMENTS AND RESULTS

To demonstrate the superior performance of WRT, we evaluate our model using both synthetic and real-world graphs, compared with other graph-partitioning methods. We firstly introduce the dataset details, then give the competitors in graph partitioning, and finally show the overall performance and ablation studies results.

### 6.1 GRAPH DATASETS

To make precise evaluation of different methods, we construct three types of graph datasets. The detailed definitions are in the following:

**Predefined-weight Graph:** In our synthetic graph data generation process, we design the structure to resemble a spider web, which consists of $N$ concentric circles, each having $M$ equally spaced nodes. The radii of circles are from 1 to $N$. Given an unweighted spider web graph, built by randomly choosing the number of circles and nodes, we randomly select a valid ring-wedge partition configuration, specifying both the number of rings and wedges. We then assign lower weights to

edges that cross different partitions and higher weights to edges within the same partition (intra-partition edges). An example of synthetic spider web graph is given in Figure 5. More details about the ranges of nodes, circles, weights etc, for generating the graphs are included in Appendix.

**Random-weight Graph:** The graph structure is the same as above, but edge weights are assigned randomly in a given range. In the random-weight graphs, models should find best partition without prior knowledge. The statistics of our training and test synthetic graphs are shown in Table 4.

**Real City Traffic Graph:** For real-world data, we utilize sub-graphs randomly extracted from a comprehensive city traffic map (Figure 5 (b)). The extracted sub-graph is always connected. For edge weights, we collect traffic data of the city during a specific time range to assess our method's ability to handle real traffic volumes effectively. The statistical information of Real City Traffic Graph can be found in Table 4.

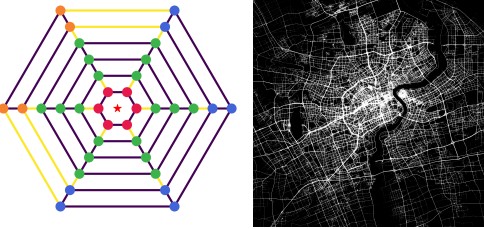

(a) Synthetic Graph    (b) City Traffic Map

Figure 5: (a) Synthetic graph: It is composed of 6 circles and 6 wedges. The edge with yellow color has lower weight, the rest edge have higher weight. The ground truth partition is composed of 2 rings and 2 wedges which nodes are in different colors. (b) Overview of Real Traffic Map, we randomly sample connected sub-graph in training and testing.

## 6.2 MODELS AND COMPARED METHODS

We compare our proposed method with the following baselines and methods.

For traditional approaches, we select: **METIS** solver that is used to partitioning graphs with balanced size. **Spectral Clustering** uses eigenvectors and k-means to perform graph partitioning.

We also propose two baselines for ring and wedge partitions: **Bruteforce** method to enumerate possible ring and wedge partitions. **Random** to randomly generate 10,000 partitions and choose the best performance one as the result.

For Reinforcement Learning based graph partitioning methods, we select two state-of-the-art methods, **ClusterNet** and **NeuroCUT**, which are introduced in Subsection 2.2.

Finally, we compare above methods with our proposed WRT and its variants. They are: **WRT** the standard Wedge-Ring Partition with two-stage training. **WRT**$_{e2e}$ directly learns Wedge-Ring Partition without two-stage training. **WRT**$_{sr}$ uses the same reward function during two training stages. **WRT**$_{nfw}$ does not freezing the wedge action network during the second training stage. **WRT**$_{nfw}$ does not performing post refinement after ring-wedge partition is generated. Results of variants are in Appendix.

## 6.3 PERFORMANCE EVALUATION

### 6.3.1 EVALUATION OF MODEL OVERALL PERFORMANCE

We show the overall performance in Table 1. We tested our model in three different types of datasets described in Section 6.1 and summerized in Table 4, with 4 or 6 partition numbers. The number of graphs used for training is $400,000$. We test the performance of different methods on 100 randomly generated graphs and report the average performance. We can find that our method always performs best compared with other methods on all datasets, showing its superior performance compared with existing methods, with the reduced ring-wedge shaped action space. Although Metis and Spectral Clustering can give graph partitions with any shape, they still cannot reach better performance compared with our proposed methods, because it is hard to find best results in such huge action space. Two basic methods, Bruteforce and Random, performs always worse compared with other methods, because they do not consider the differences of edge weights, and only do random partitioning.

### 6.3.2 EVALUATION OF MODEL TRANSFER PERFORMANCE

We train model on those three types of graphs with number of nodes (N=100) on each circle and conduct graph partition transfer learning experiments on the graphs with number of nodes $N = 50$

Table 1: Performance on Predefined-weight, Random-weight, and City Traffic Graphs by Normalized cut. Lower values indicate better performance. Best value (bold), 2nd best (underline).

| Method | Predefined-weight | | | | Random-weight | | | | City Traffic | | | |
|---|---|---|---|---|---|---|---|---|---|---|---|---|
| | 4 Part. | | 6 Part. | | 4 Part. | | 6 Part. | | 4 Part. | | 6 Part. | |
| | 50 | 100 | 50 | 100 | 50 | 100 | 50 | 100 | 50 | 100 | 50 | 100 |
| Metis | .069 | .036 | .097 | .053 | .065 | .033 | .094 | .049 | .245 | .162 | .383 | .304 |
| Spec. Clust. | .065 | .036 | .099 | .053 | .079 | .041 | .101 | .053 | .384 | .218 | .652 | .843 |
| Bruteforce | .070 | .036 | .106 | .054 | .070 | .036 | .107 | .054 | .361 | .237 | .615 | .457 |
| Random | .076 | .040 | .144 | .074 | .080 | .041 | .142 | .072 | .209 | .095 | .512 | .341 |
| NeuroCut | .059 | .032 | .086 | .046 | .064 | .033 | .093 | .049 | .192 | .078 | .348 | .226 |
| ClusterNet | .078 | .043 | .106 | .070 | .093 | .043 | .120 | .083 | .507 | .261 | .837 | .747 |
| WRT | **.042** | **.021** | **.062** | **.032** | **.057** | **.029** | **.081** | **.041** | **.174** | **.060** | **.317** | **.182** |

Table 2: Transfer performance measured by Normalized Cut. Methods that do not support transfer or unable to perform results are excluded. Models are trained on 100 nodes and tested on 50 or 200 nodes. Best value (bold) and 2nd best value (underline).

| Partition | Predefined-weight | | | | Random-weight | | | | City Traffic | | | |
|---|---|---|---|---|---|---|---|---|---|---|---|---|
| | 4 Part. | | 6 Part. | | 4 Part. | | 6 Part. | | 4 Part. | | 6 Part. | |
| Nodes | 50 | 200 | 50 | 200 | 50 | 200 | 50 | 200 | 50 | 200 | 50 | 200 |
| METIS | .069 | .019 | .097 | .027 | .065 | .016 | .094 | .024 | .245 | .048 | .383 | .086 |
| Bruteforce | .070 | .018 | .106 | .028 | .070 | .018 | .107 | .028 | .361 | .175 | .615 | .311 |
| Random | .076 | .021 | .144 | .037 | .080 | .021 | .142 | .037 | .209 | .512 | .512 | .212 |
| WRT | **.052** | **.013** | **.066** | **.017** | **.061** | **.016** | **.087** | **.022** | **.158** | **.023** | **.323** | **.085** |

and $N = 200$ without fine-tuning. The result in table 2 shows that our model has great generalizability, when trained on certain size of graphs, it is able to apply on different size, regardless of node number becomes bigger or smaller.

### 6.3.3 RINGNESS AND WEDGENESS EVALUATION

Table 3 shows the quantification results of Ringness and Wedgeness on City Traffic Graphs. We can find that WRT also reaches the best Ringness and Wedgeness compared with other methods.

Table 3: Ringness and Wedgeness Evaluation of different methods, higher is better.

| | METIS | Spec.Clust. | NeuroCUT | ClusterNet | WRT |
|---|---|---|---|---|---|
| Ringness | 0.871 | 0.776 | 0.840 | 0.854 | **0.929** |
| Wedgeness | 0.587 | 0.810 | 0.621 | 0.820 | **0.876** |

## 7 CONCLUSION AND FUTURE WORK

In this paper, we have demonstrated the efficacy of using Reinforcement Learning for solving a special form of the normalized cut problem on weighted graphs, an area where traditional methods like METIS fall short, and eixsting RL based graph partitioning methods also cannot perform well when the initial partition generated by METIS is not good enough. Inspired by urban road network construction, we propose to make ring and wedge partition directly on graphs. By introducing the simplified partitioning strategy involving ring-shaped and wedge-shaped cuts, our approach leverages RL and Transformers to effectively learn and optimize the partitioning process. The two-stage training methodology ensures stability and scalability, enabling our algorithm to handle both small and large graphs efficiently. Our experimental results highlight the superiority of our method over baseline algorithms, showcasing its potential for real-world applications. Our proposed method focus on minimizing normalized cut of planar graphs, future work will focus on extend existing methods to non-planar graphs, and find better post-process methods to further improve the final performance.

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

## A Statistics and Hyper Parameters

Table 4: Statistics of datasets and hyper parameters of model.

| Type | Parameter | Values | Description |
|---|---|---|---|
| Synthetic | Nodes | {50, 100} | Nodes on each circle |
| | Circles | same as Nodes | Number of concentric rings |
| | Low Weight | {2, 4, 6} | Intra-partition edge weights |
| | High Weight | {10, 15, 20} | Inter-partition edge weights |
| | Random Weight | Uniform(1, 10) | Edge weights for random |
| Real | Nodes | {50, 100} | Nodes on each graph |
| | Edge Weight | [1, 372732] | Edge weights |
| Hyper Parameters | Partitions | {4,6} | Number of partitions |
| | Hidden Size | 64 | Hidden size of Transformer |
| | Layer Number | 3 | Transformer layer number |
| | Learning Rate | 1e-3 | Learning rate |
| | Batch Size | 256 | Batch size |
| | Discount Factor | 0.9 | Discount factor in RL |
| | Training Step | 400,000 | Steps for training |
| | Test Number | 100 | Test graph number |
| | Sample Number | 10 | Sampled partition number |

## B Ablation Studies

We give ablation studies in the following to show the effectiveness of proposed methods. The performance of ablation models are shown in Table 5 and 6.

### B.1 Two-stage Training and Testing

In Section 5.5, we propose multi-stage traininig and testing strategies. In training, we propose to train the wedge partition model firstly, and randomly select ring partitions. The radius are uniformly selected from 0 to 80% of maximum radius. After wedge partition model is trained, we re-initialize the critic projection header of wedge model, and fix the other parts of wedge model to train the ring model part. We show the performance without two stage training as $\text{WRT}_{e2e}$. From Table 5 and Table 6, we can find that without two stage training, the model is not able to converge, because a bad policy of either ring or wedge will affect the learning process of each other, and make the model hard to converge.

### B.2 Different Baseline Function in Two-stage

As mentioned in Section 5.5, in training wedge partition, we change the reward function from global Normalized Cut to the Normalized Cut that only considering wedge partitions. This avoids the impact of poor ring partition selection, as ring partition is performed by a random policy, and may give poor partitions. For example, if the random policy selects a very small radius, the normalized cut of circle partitions will be very big, which makes the reward received from different wedge partition identical. We show the performance when the reward function keeps same, i.e. always considering the normalized cut of ring partition in two stage training, as $\text{WRT}_{sr}$. In Table 5 and 6, we can find that their performance is worse than WRT, because their wedge partitioning policies are not strong enough. As reward function will change in two-stage training, we will re-initialize the critic net of wedge model in the second stage, as mentioned before.

### B.3 Fix Wedge Partition Policy

In the second training stage, we fix the wedge model to avoid changing the policy. $\text{WRT}_{nfw}$ shows the performance when wedge partition policy is not fixed. We can find that the performance decreases if wedge partition policy is not fixed, and leads to bad policy in several test cases. This is because if we allow the action net change, it may forget learned policy before a valid policy has

Table 5: Performance comparison on Predefined-weight, Random-weight, and City Traffic Graphs (Normalized cut). Lower values indicate better performance.

| Method | Predefined-weight | | | | Random-weight | | | | City Traffic | | | |
| --- | --- | --- | --- | --- | --- | --- | --- | --- | --- | --- | --- | --- |
| | 4 Part. | | 6 Part. | | 4 Part. | | 6 Part. | | 4 Part. | | 6 Part. | |
| | 50 | 100 | 50 | 100 | 50 | 100 | 50 | 100 | 50 | 100 | 50 | 100 |
| Metis | .069 | .036 | .097 | .053 | .065 | .049 | .094 | .049 | .245 | .162 | .383 | .304 |
| Spec. Clust. | .065 | .036 | .099 | .053 | .079 | .053 | .101 | .053 | .384 | .218 | .652 | .843 |
| Bruteforce | .070 | .036 | .106 | .054 | .070 | .054 | .107 | .054 | .361 | .237 | .615 | .457 |
| Random | .076 | .040 | .144 | .074 | .080 | .072 | .142 | .072 | .209 | .095 | .512 | .341 |
| NeuroCut | .059 | .032 | .086 | .046 | .064 | .033 | .093 | .049 | .192 | .078 | .348 | .226 |
| ClusterNet | .078 | .043 | .106 | .070 | .093 | .043 | .120 | .083 | .507 | .261 | .837 | .747 |
| $\text{WRT}_{sr}$ | .063 | .276 | .065 | .032 | .159 | .044 | .091 | .046 | .646 | .473 | .792 | .612 |
| $\text{WRT}_{e2e}$ | .105 | .053 | .123 | .063 | .112 | .055 | .131 | .069 | .683 | .478 | .783 | .678 |
| $\text{WRT}_{o}$ | .042 | .021 | .062 | .032 | .057 | .029 | .081 | .041 | .209 | .071 | .419 | .271 |
| $\text{WRT}_{nfw}$ | .046 | .023 | .065 | .033 | .057 | .029 | .082 | .041 | .175 | .060 | .328 | .187 |
| WRT | .042 | .021 | .062 | .032 | .057 | .029 | .081 | .041 | .174 | .060 | .317 | .182 |

Table 6: Transfer performance measured by Normalized Cut. Methods that do not support transfer or unable to perform results are excluded. Models are trained on 100 nodes and tested on 50 or 200 nodes.

| Partition | Predefined-weight | | | | Random-weight | | | | City Traffic | | | |
| --- | --- | --- | --- | --- | --- | --- | --- | --- | --- | --- | --- | --- |
| | 4 Part. | | 6 Part. | | 4 Part. | | 6 Part. | | 4 Part. | | 6 Part. | |
| Nodes | 50 | 200 | 50 | 200 | 50 | 200 | 50 | 200 | 50 | 200 | 50 | 200 |
| METIS | .069 | .019 | .097 | .027 | .065 | .016 | .094 | .024 | .245 | .048 | .383 | .086 |
| Bruteforce | .070 | .018 | .106 | .028 | .070 | .018 | .107 | .028 | .361 | .175 | .615 | .311 |
| Random | .076 | .021 | .144 | .037 | .080 | .021 | .142 | .037 | .209 | .512 | .512 | .212 |
| $\text{WRT}_{sr}$ | .219 | .201 | .068 | .018 | .085 | .023 | .092 | .024 | .664 | .305 | .831 | .224 |
| $\text{WRT}_{e2e}$ | .103 | .027 | .107 | .027 | .107 | .028 | .123 | .033 | .645 | .327 | .863 | .442 |
| $\text{WRT}_{o}$ | .052 | .013 | .066 | .016 | .061 | .016 | .087 | .022 | .182 | .031 | .472 | .014 |
| $\text{WRT}_{nfw}$ | .053 | .013 | .066 | .017 | .061 | .016 | .087 | .104 | .150 | .023 | .327 | .090 |
| WRT | .052 | .013 | .066 | .017 | .061 | .016 | .087 | .022 | .158 | .023 | .323 | .085 |

learned by ring partition, and leads to worse performance and instability during the training. The reward curve during training and testing, which is shown in Figure 6, also supports the conclusion. It has been observed that not fixing the action net results in lower and more unstable rewards for the model during training. Moreover, the performance during testing tends to become more variable and does not show further improvements as training progresses.

## B.4 POST REFINEMENT

We perform post refinement after performing the ring and wedge partition, which splits existing partition result by the connectivity of nodes, then reconstruct new partitions by combining the partition which has biggest Normalized Cut with its adjacent partitions. As ring and wedge partitions on synthetic graphs are always connected, this post refinement will not change the performance of WRT on synthetic dataset. However, in real dataset, sometimes the graph shape is not compatible to ring and wedge partition, and the results may not good enough. With post refinement, we can further decrease the Normalized Cut on such situation. In Table 5, we show the performance improvements with post refinement on real dataset, the normalized cut is decreased 22.4% on average.

## B.5 GRAPH CENTER SELECTION

We conducted experiments on the test set of City Traffic graphs with 50 nodes, which contains 100 graphs. Based on the maximum aspect ratio of the graphs, we offset the centroid by a distance of up to 5% and recalculated the results of Normalized Cut. For better comparison, we normalized the results using the Normalized Cut from the unoffset scenario. A normalized value closer to zero

Table 7: Performance comparison on City Traffic Graphs (Normalized cut). Lower values indicate better performance.

| Method | City Traffic | | | |
| --- | --- | --- | --- | --- |
| | 4 Part. | | 6 Part. | |
| | 50 | 100 | 50 | 100 |
| GPSGNN | 0.696 | 0.521 | 0.821 | 0.711 |
| Transformer | 0.186 | 0.122 | 0.360 | 0.225 |
| WRT w/o PAMHA | 0.169 | 0.075 | 0.374 | 0.193 |
| WRT | 0.174 | 0.060 | 0.317 | 0.182 |

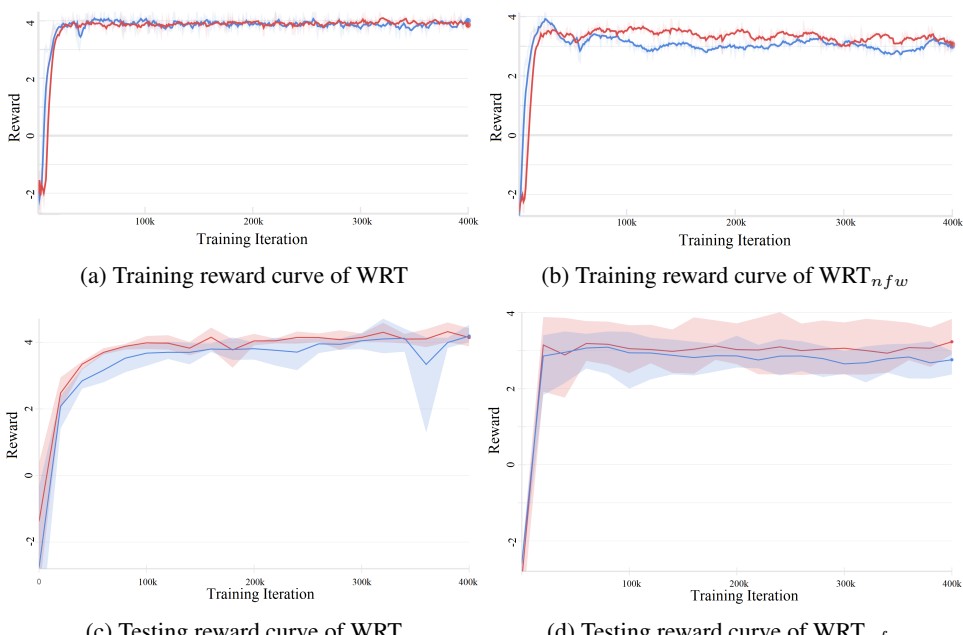

(a) Training reward curve of WRT

(b) Training reward curve of $\text{WRT}_{nfw}$

(c) Testing reward curve of WRT

(d) Testing reward curve of $\text{WRT}_{nfw}$

Figure 6: Reward curves during training and testing of Predefined-weight Graph. Red is with 50 node number and blue is with 100 node number. We individually perform 4 tests for each checkpoint, using the curve to represent the average test results, while the shaded area indicates the maximum and minimum values observed during the tests.

indicates better performance, with a value of 1 signifying that the results are the same as in the unoffset case.

Figure 7 (left) illustrates the results for various offsets from the centroid. We observe that any offset from the centroid results in a worse performance, and with greater offsets correlating to a more significant decline.

In Figure 7 (right), we present the histogram of results across all the aforementioned offsets and graphs. We find that in nearly half of the cases where offsets were applied, the resulting errors remained within 5%. Furthermore, applying offsets tends to lead to worse outcomes more frequently. Thus, in this paper, we opted to use the centroid as the center of the graph. Figure 7 (right) also shows that in approximately 15% of cases, offsetting the centroid yielded improvements of over 10%. In the future, we can propose a more effective strategy for centroid selection to enhance the algorithm's performance.

## B.6 EFFECTIVENESS OF GRAPH TRANSFORMATION, WRT AND PAMHA

We show the effectiveness of our proposed Graph Transformation, WRT and PAMHA in Table 7. The methods are:

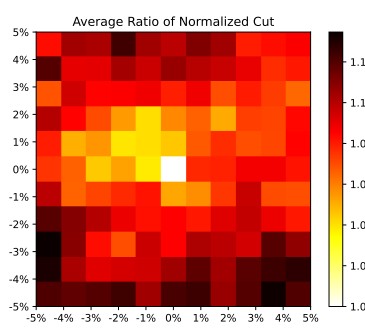 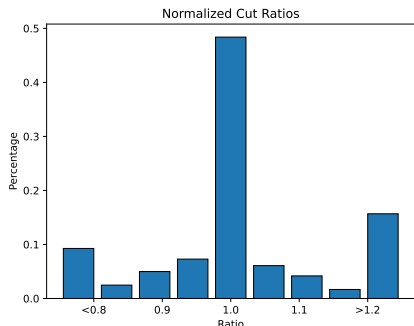

Figure 7: Heatmap and histogram of Normalized Cut when applying offsets to the center. Values are normalized by Normalized Cut of unoffset center. Lower value is better.

- GPSGNN [1]. It is an advanced Graph Neural Network, which combines GNN and Transformers to both collect local and global information. This represents the initial performance of GNNs and Transformers in solving Normalized Cut problem with Ring and Wedge Partition.

- Transformer. We perform Ring Transformation and Wedge Transformation of the graph. Then use node coordinates as positional embeddings, and edge weights as attention masks. Then we use the vanilla Transformer.

- WRT w/o PAMHA. The same structure as WRT, except PAMHA is replaced with normal Multi-Head Attention.

We demonstrate the effectiveness of our proposed Graph Transformation methods, WRT and PAMHA, as presented in Table 7. The methods evaluated are as follows:

- GPSGNN [1]: It integrates GNNs and Transformers to effectively capture both local and global information. It serves as the benchmark for evaluating the performance of GNNs and Transformers in addressing the Normalized Cut problem using Ring and Wedge Partition techniques.

- Transformer: We implement Ring and Wedge Transformations on the graph, utilizing node coordinates as positional embeddings and edge weights as attention masks, followed by the application of a standard Transformer model.

- WRT w/o PAMHA: This configuration maintains the same architecture as WRT but substitutes PAMHA with conventional Multi-Head Attention.

From the results, we observe that GPSGNN struggles to produce meaningful results in City Traffic. In contrast, the Transformer model demonstrates significantly improved outcomes when compared to GPSGNN. This finding reinforces the notion that constraining the action space of RL agents based on domain knowledge is essential for enhancing performance. Next, we integrated the WRT structure, which facilitates data pre-calculation, and observed an increase in performance. This indicates that the pre-calculation module is beneficial in addressing the problem. Finally, by incorporating PAMHA into the Transformer framework, we achieved the full WRT architecture, which exhibited superior performance relative to other variants.

[1] Ladislav Rampášek, Mikhail Galkin, Vijay Prakash Dwivedi, Anh Tuan Luu, Guy Wolf, and Dominique Beaini. Recipe for a General, Powerful, Scalable Graph Transformer. Advances in Neural Information Processing Systems, 35, 2022.

## B.7 PERFORMANCE OF OTHER METHODS

We incorporate additional comparative methods Analyzed on City Traffic Graph, which include:

- DMon [2]: A neural attributed graph clustering method designed to effectively handle complex graph structures.

Table 8: Performance comparison on City Traffic Graphs (Normalized cut). Lower values indicate better performance.

| Method | City Traffic | | | |
| | 4 Part. | | 6 Part. | |
| | 50 | 100 | 50 | 100 |
| --- | --- | --- | --- | --- |
| DMon | 0.998 | 1.000 | 1.000 | 1.000 |
| MinCutPool | 0.549 | 0.365 | 0.864 | 0.634 |
| Ortho | 0.924 | 0.892 | 0.997 | 0.982 |
| WRT | 0.174 | 0.060 | 0.317 | 0.182 |

Table 9: Comparison of Hidden State and Learning Rate for Different Methods.

| | Hidden State | Learning Rate |
| --- | --- | --- |
| Search Range | 16, 32, 64, 128, 256 | 1e-4, 3e-4, 1e-3, 3e-3, 1e-2 |
| Dmon | 128 | 1e-4 |
| MinCutPool | 32 | 1e-3 |
| Ortho | 256 | 1e-2 |

- MinCutPool [3]: This method focuses on optimizing the normalized cut criterion while incorporating an orthogonality regularizer to mitigate unbalanced clustering outcomes.
- Ortho [2]: This refers to the orthogonality regularizer that is utilized in both DMon and MinCutPool, ensuring greater balance in the clustering process.

All models were trained using the same settings as WRT. We conducted a grid search on the hyperparameters of the three aforementioned methods and selected the optimal combination of hyperparameters to train on other datasets. The search range and the selected hyperparameters are detailed in Table 9. The results are summarized in Table 8. From the findings, it is evident that Dmon fails to effectively learn the partition strategy, resulting in most outcomes being invalid (with Normalized Cut values of 1). Ortho performs slightly better but still tends to yield unbalanced results. In contrast, MinCutPool demonstrates a significant improvement over the previous methods; however, it still exhibits a considerable range compared to our proposed WRT.

[2] Anton Tsitsulin, John Palowitch, Bryan Perozzi, and Emmanuel Müller. 2023. Graph clustering with graph neural networks. Journal of Machine Learning Research 24, 127 (2023), 1–21.
[3] Filippo Maria Bianchi, Daniele Grattarola, and Cesare Alippi. 2020. Spectral clustering with graph neural networks for graph pooling. In International conference on machine learning. PMLR, 874–883.

## C    DETAIL OF THE MODEL PIPELINE

We detail the model transformation formulas below. Since the processes for ring and wedge partitioning are similar, we focus on the wedge partitioning pipeline and note the differences later.

Let $G$ be the input graph and $P$ the current partition. We apply Wedge Transformation to $G$ to obtain a linear graph $G'$. Define $n_i$ as the $i$-th node on this line, with $n$ candidate actions. Action $a_i$ corresponds to selecting the radius of the ring partition between $n_i$ and $n_{i+1}$. The input embedding is constructed as follows:

$$\boldsymbol{X}_i = \text{Linear}_{CW}(\text{Cut}_i \oplus \text{PS}_i) + \text{Pos}_{i/|N|} \to \mathbb{R}^d,$$

where $\text{Linear}_{CW}$ is a linear transformation, $\text{Cut}_i$ is the Cut Weight between $n_i$ and $n_{i+1}$, $\text{PS}_i$ is the Partition Selection for $n_i$, and $\text{Pos}_{i/|N|}$ is the positional embedding scaled based on the total number of nodes.

In WRT, we derive the attention masks $M^P$ and $M^V$ as follows:

$$M_{i,j}^P = \begin{cases} 0 & \text{if } i \text{ and } j \text{ are in the same partition} \\ -\infty & \text{otherwise} \end{cases},$$

$$M_{i,j}^V = \text{Linear}_V(V_{i,j}).$$

(5)

When nodes $i$ and $j$ are in different partitions, the attention weight is set to $-\infty$ to prevent their influence on each other. Denote $H_i^0 = X_i$; the $t$-th hidden states $H_i^t$ are computed as follows:

$$\boldsymbol{Q}^t, \boldsymbol{K}^t, \boldsymbol{V}^t = \text{Linear}_{\text{Att}}(\boldsymbol{H}^t), \tag{6}$$

$$\boldsymbol{O}^t = \text{Softmax}(\boldsymbol{Q}\boldsymbol{K}^t + \boldsymbol{M}^{\boldsymbol{V}} + \boldsymbol{M}^{\boldsymbol{P}}). \tag{7}$$

The output for each node at the $t$-th layer is:

$$Y_i^t = \text{LayerNorm}\left(\sum_{k=1}^{N} O_{i,k}^t V_k / \sqrt{d} + H_i^t\right), \tag{8}$$

$$H_i^{t+1} = \text{LayerNorm}(Y_i^t + \text{FFN}(Y_i^t)). \tag{9}$$

The Transformer has $T$ layers, with $\boldsymbol{E} = \boldsymbol{H}^T$ as the output embeddings.

In the PPO module, we calculate action probabilities and predicted values using $\boldsymbol{E}$:

$$\text{Logit}_i = \text{Linear}_A(E_i) \to \mathbb{R}, \tag{10}$$

$$\textbf{Prob} = \text{Softmax}(\text{Logit}), \tag{11}$$

$$\text{PredictedValue} = \text{Linear}_{PV}(\text{Attention}(\boldsymbol{E})) \to \mathbb{R}. \tag{12}$$

We sample actions from the probability distribution and train the model using rewards and predicted values.

For ring partitioning, two key differences arise. First, we employ Ring Transformation on the graph. Second, the positional embedding is 2-dimensional, reflecting the adjacency of the first and last nodes. Specifically, for a circular node with coordinates $(x, y)$, we use:

$$PE = \text{Linear}_{2D}(x \oplus y) \to \mathbb{R}^d$$

to generate the positional embedding.

## D    DEFINITION OF RINGNESS AND WEDGENESS

We propose the Ringness and Wedgeness to evaluate whether a partition is close to the ring shape or wedge shape. We expect a typical Ring and Wedge partition will have the highest Ringness and Wedgeness.

For partition $p_i \in P$, we define the partition range $pr_i = \{\min(\boldsymbol{r}_i), \max(\boldsymbol{r}_i)\}$, partition angle $pa_i = \{\min(\boldsymbol{a}_i), \max(\boldsymbol{a}_i)\}$, where $\boldsymbol{r}_i$ and $\boldsymbol{a}_i$ are polar coordinates of nodes that belongs to $p_i$.

Then we define Ringness for a partition $R_P(r) = |\{r \in pr_i\}|$, which means that how many partitions cover the radius $r$. For a pure ring partition, as different partitions will never overlap within their radius, $R_P(r)$ will be always 1; and for a Ring and Wedge partition, $R_P(r)$ is always 1, except the out-most wedge part, is the wedge partition number $k_w$.

For Wedgeness, we define $W_P(r) = \sum_{r \in pr_i} |pa_i|$, where $|pa_i|$ is the angle range of $pa_i$. For radius $r$, we only consider the partition that covers the selected range, and we sum up the angles covered by these partitions. The angle should equal or greater than $2\pi$, as the graph is fully partitioned by $P$. If a partition is a pure wedge partition, for any $r$, the Wedgeness should be exactly $2\pi$, because partition will never cover each other in any place. For Ring and Wedge Partition, if $r$ is in Wedge Partition part, the conclusion remains same as above; for Ring Partition part, only one partition is selected, and the Wedgeness is also $2\pi$.

To represent Ringness and Wedgeness more clearly, we calculate the quantification metrics for them based on the following formula:

$$\boldsymbol{W}_P = \left(Z(P) - \min_{0 \le k \le \max(\boldsymbol{r})}\left(\int_{i=0}^{k} W_P(i) + \int_{i=k}^{\max(\boldsymbol{r})} (\max(W) - W_P(i))\right)\right) / Z(P) \tag{13}$$

$$\boldsymbol{R}_P = \frac{2\pi}{\max_r R_P(r)} \tag{14}$$

$$f(x) = \begin{cases} 1 & \text{if} \quad 0 \le x \le k \\ \max(W) & \text{if} \quad k < x \le max(\boldsymbol{r}) \end{cases} \tag{15}$$

Here $Z(P) = 0.5 \max(\boldsymbol{r}) \cdot \max(W)$ is the normalization factor. We use a piecewise function $f$ to approximate $W_P$, and provide $\boldsymbol{W}_P$ based on the difference between $W_P$ and $f$. For $\boldsymbol{R}_P$, we select the maximum of $R_p$. Both $\boldsymbol{W}_P$ and $\boldsymbol{R}_P$ is scaled to $[0, 1]$, and the higher means the better.

# E  PROOFS OF CHEEGER BOUNDS

## E.1  PROOF OF PROPOSITION 1

In this section we will provide all the details of the proof of Proposition 1. First we recall some background definitions and results.

Let $G = (V, E)$ be an undirected graph with $|V| = n$. Let $D$ be the diagonal matrix with the node degrees on the diagonal and let $L = D^{-\frac{1}{2}}(D - A)D^{-\frac{1}{2}}$ be the normalized Laplacian of $G^1$, where $A$ is the adjacency matrix of $G$. The matrix $L$ is positive semi-definite with eigenvalues

$$0 = \lambda_1 \leq \lambda_2 \leq \ldots \leq \lambda_n. \tag{16}$$

For a subset $S \subseteq V$ define

$$\phi_G(S) = \frac{Cut(S, S^c)}{Volume(S)} \tag{17}$$

and, for $1 \leq k \leq n$, we define the *Cheeger constants*

$$\rho_G(k) = \min_{\substack{S_1, \ldots, S_k \\ \text{partition of V}}} \max_{1 \leq i \leq k} \phi_G(S_i). \tag{18}$$

It is known that, for $k = 2$, the following inequalities hold Chung (1997)

$$\frac{\lambda_2}{2} \leq \rho_G(2) \leq \sqrt{2\lambda_2}. \tag{19}$$

Analogous inequalities were proved in Lee et al. (2014) for every $1 \leq k \leq n$

$$\frac{\lambda_k}{2} \leq \rho_G(k) \leq \mathcal{O}(k^2)\sqrt{\lambda_k}. \tag{20}$$

If $G$ is planar, then the right-side inequality can be improved and reads

$$\rho_G(k) \leq \mathcal{O}(\sqrt{\lambda_{2k}}). \tag{21}$$

Now let $G_{N,r} = (V, E)$ be an undirected spider web graph with $r$ rings and $N$ points for each ring. This is exactly the cartesian product of a circle graph and a path graph with $N$ and $r$ vertices respectively. Note that the "center" is not included in this type of graphs. Define the following custom Cheeger constants:

$$\varphi_{N,r}(k) = \min_{\substack{S_1, \ldots, S_k \\ \text{wedge partition of V}}} \max_{1 \leq i \leq k} \phi_{G_{N,r}}(S_i) \tag{22}$$

$$\psi_{N,r}(k) = \min_{\substack{S_1, \ldots, S_k \\ \text{ring partition of V}}} \max_{1 \leq i \leq k} \phi_{G_{N,r}}(S_i). \tag{23}$$

For an illustration of wedge and ring partitions see Figure 8. From now on we will assume $G = G_{N,r}$ to be a spider web graph with $r$ rings and $N$ points for each ring. We can compute bounds on $\varphi_{N,r}(k)$ and $\psi_{N,r}(k)$.

**Lemma 1** *Given a spider-web graph $G_{N,r}$ the wedge and ring Cheeger constants can be bounded as follows*

$$\varphi_{N,r}(k) \leq \frac{r}{\lfloor \frac{N}{k} \rfloor (2r - 1)}, \quad \psi_{N,r}(k) \leq \frac{1}{2\lfloor \frac{r}{k} \rfloor}. \tag{24}$$

**Proof 1** *The strategy will be to choose suited partitions for which it is possible to compute explicitly the cuts and the volumes. We start from the wedge Cheeger constant $\varphi_{N,r}(k)$. Given a wedge partition $S_1, \ldots, S_k$ each subset $S_i$ has cut exactly $2r$. Moreover, we assume that the $S_i$'s are*

---

[1]For the sake of simplicity, often it will be called just Laplacian.

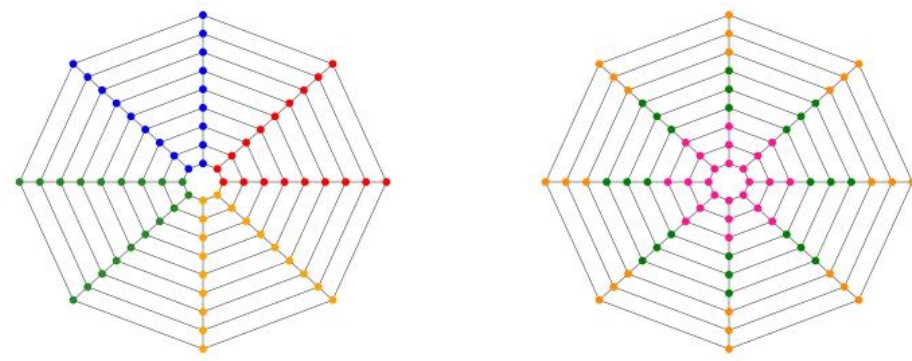

Figure 8: Examples of $k = 4$ wedge (left) and $k = 3$ ring (right) partitions.

*maximally symmetric, meaning that the $S_i$'s all have $\lfloor \frac{N}{k} \rfloor$ or $\lfloor \frac{N}{k} \rfloor + 1$ nodes in each ring. These observation read*

$$\varphi_{N,r}(k) \leq \max_{1 \leq i \leq k} \frac{2r}{Volume(S_i)}$$
$$= \frac{2r}{\min_{1 \leq i \leq k} Volume(S_i)}. \tag{25}$$

*The wedge subset with minimum volume is given by one that has $\lfloor \frac{N}{k} \rfloor$ nodes in each ring, hence*

$$\varphi_{N,r}(k) \leq \frac{2r}{\underbrace{4}_{\substack{degree\ of \\ inner\ ring \\ nodes}} \underbrace{(r-2)}_{\substack{number\ of \\ inner\ rings}} \underbrace{\lfloor \frac{N}{k} \rfloor}_{\substack{number\ of \\ points\ in \\ each\ ring}} + \underbrace{3}_{\substack{degree\ of \\ outer\ ring \\ nodes}} \underbrace{2}_{\substack{number\ of \\ outer\ rings}} \underbrace{\lfloor \frac{N}{k} \rfloor}_{\substack{number\ of \\ points\ in \\ each\ ring}}}$$
$$= \frac{r}{\lfloor \frac{N}{k} \rfloor (2r - 1)}. \tag{26}$$

*For the ring Cheeger constant the setting is more complicated since different subsets might have different cut, in contrast with the case of wedge partitions. Given a ring partition $S_1, \ldots, S_k$ which is maximally symmetric, i.e., all the $S_i$'s have $\lfloor \frac{r}{k} \rfloor$ or $\lfloor \frac{r}{k} \rfloor + 1$ nodes in each ring, we order the $S_i$'s from the center to the outermost ring. Note that $S_1$ has some nodes with degree 3 while $S_2$ has all nodes with degree 4 for $k > 2$. For $k_¿$ 2, we consider two cases:*

- *$k$ divides $r$. In this case we only need to compare $S_1$ and $S_2$. It holds that*

$$\phi_G(S_1) = \frac{N}{4N(\frac{r}{k} - 1) + 3N} = \frac{1}{4\frac{r}{k} - 1} \tag{27}$$

$$\phi_G(S_2) = \frac{2N}{4N\frac{r}{k}} = \frac{1}{2\frac{r}{k}}, \tag{28}$$

  *since $S_1$ and $S_2$ have cut $N$ and $2N$ respectively. Thus, $\phi_G(S_2) \geq \phi_G(S_2)$ which implies $\psi_{N,r}(k) \leq \frac{1}{2\frac{r}{k}}$.*

- *$k$ does not divide $r$. In this case, we assume $S_1$ has $\lfloor \frac{r}{k} \rfloor + 1$ nodes and $S_2$ has $\lfloor \frac{r}{k} \rfloor$ nodes. Then*

$$\phi_G(S_1) = \frac{N}{4N\lfloor \frac{r}{k} \rfloor + 3N} = \frac{1}{4\lfloor \frac{r}{k} \rfloor + 3} \tag{29}$$

$$\phi_G(S_2) = \frac{2N}{4N\lfloor \frac{r}{k} \rfloor} = \frac{1}{2\lfloor \frac{r}{k} \rfloor}. \tag{30}$$

  *Thus, $\phi_G(S_2) \geq \phi_G(S_2)$ which implies $\psi_{N,r}(k) \leq \frac{1}{2\lfloor \frac{r}{k} \rfloor}$.*

For $k = 2$, if $k$ divides $r$, then $\phi_G(S_1) = \phi_G(S_2) = \frac{1}{4\frac{r}{k}-1} \leq \frac{1}{2\frac{r}{k}}$. If $k$ does not divide $r$, then if $S_1$ has $\lfloor \frac{r}{k} \rfloor + 1$ nodes and $S_2$ has $\lfloor \frac{r}{k} \rfloor$ nodes, we have $\phi_G(S_1) = \frac{1}{4\lfloor \frac{r}{k} \rfloor+3}$ and $\phi_G(S_2) = \frac{1}{4\lfloor \frac{r}{k} \rfloor-1} \leq \frac{1}{2\lfloor \frac{r}{k} \rfloor}$. Putting together the above inequalities, we get that $\psi_{N,r}(k) \leq \frac{1}{2\lfloor \frac{r}{k} \rfloor}$.

In the next sections we will provide the proof details for the two bounds in Proposition 1. We start from the case of wedge partitions.

### E.1.1  WEDGE PARTITIONS

We will prove the bound on wedge Cheeger constants in terms of the eigenvalues of the circle graph with $N$ vertices $C_N$. We will consider only the case of $k > 1$ since the first eigenvalues is always $0$ and spider web graphs are connected. First we recall that the eigenvalues of $C_N$ are

$$1 - \cos\left(\frac{2\pi k}{N}\right), \quad 0 \leq k \leq N - 1, \tag{31}$$

see Chung (1997). In particular, we have the following result.

**Lemma 2** *Let $C_N$ be the circle graph with $N$ vertices. Then the $k$-th eigenvalues of the normalized Laplacian of $C_N$ is given by*

$$\lambda_k^C = 1 - \cos\left(\frac{2\pi\lfloor \frac{k}{2} \rfloor}{N}\right), \quad 1 \leq k \leq N. \tag{32}$$

**Proof 2** *If we order the values of $\left\{1 - \cos\left(\frac{2\pi(k-1)}{N}\right)\right\}_{k=1}^{N}$ we notice that*

$$\lambda_k^C = \begin{cases} f(\frac{k}{2}) & if \quad k \in 2\mathbb{Z} \\ f(\frac{k-1}{2}) & if \quad k \notin 2\mathbb{Z} \end{cases} \tag{33}$$

*where $f(k) = 1 - \cos\left(\frac{2\pi k}{N}\right)$. Writing together the two pieces in equation 33 we get*

$$\lambda_k^C = 1 - \cos\left(\frac{2\pi\lfloor \frac{k}{2} \rfloor}{N}\right), \quad 1 \leq k \leq N. \tag{34}$$

Now we will prove some inequalities that together will build the final wedge Cheeger inequality.

**Lemma 3** $\pi\lfloor \frac{k}{2} \rfloor \frac{1}{N} \leq \frac{\pi}{2}$, *for* $2 \leq k \leq N$.

**Proof 3** *Since $k \leq N$ we have the following inequality*

$$\pi\lfloor \frac{k}{2} \rfloor \frac{1}{N} \leq \pi\lfloor \frac{N}{2} \rfloor \frac{1}{N} \begin{cases} = \frac{\pi}{2} & if \quad k \in 2\mathbb{Z} \\ = \pi\frac{N-1}{2}\frac{1}{N} \leq \frac{\pi}{2} & if \quad k \notin 2\mathbb{Z} \end{cases} \tag{35}$$

**Lemma 4** $2\lfloor \frac{k}{2} \rfloor \geq \frac{k}{2}$, *for* $2 \leq k \leq N$.

**Proof 4** *If $k$ is even then $2\lfloor \frac{k}{2} \rfloor = 2\frac{k}{2} \geq \frac{k}{2}$. If $k$ is odd, then $2\lfloor \frac{k}{2} \rfloor = 2\frac{k-1}{2} = k - 1 \geq \frac{k}{2}$ for $2 \leq k \leq N$.*

**Lemma 5** $\sqrt{\lambda_k^C} \geq \frac{\sqrt{2}}{4}\frac{1}{\lfloor \frac{N}{k} \rfloor}$, *for* $2 \leq k \leq N$.

**Proof 5** *It holds that*

$$\sqrt{\lambda_k^C} = \sqrt{1 - \cos\left(\frac{2\pi\lfloor\frac{k}{2}\rfloor}{N}\right)} \tag{36}$$

$$= \sqrt{2}\sin\left(\frac{\pi\lfloor\frac{k}{2}\rfloor}{N}\right) \tag{37}$$

$$\geq \sqrt{2}\frac{2}{\pi}\left(\frac{\pi\lfloor\frac{k}{2}\rfloor}{N}\right) \tag{38}$$

$$= 2\sqrt{2}\lfloor\frac{k}{2}\rfloor\frac{1}{N} \tag{39}$$

$$\geq \sqrt{2}\frac{k}{2}\frac{1}{N} \tag{40}$$

$$\geq \frac{\sqrt{2}}{2}\frac{1}{\lfloor\frac{N}{k}\rfloor + 1} \tag{41}$$

$$\geq \frac{\sqrt{2}}{2}\frac{1}{2\lfloor\frac{N}{k}\rfloor} \tag{42}$$

$$= \frac{\sqrt{2}}{4}\frac{1}{\lfloor\frac{N}{k}\rfloor} \tag{43}$$

*where equation 36 follows from Lemma 2, equation 37 follows from the fact that $\cos(2x) = 1 - 2\sin^2(x)$, equation 38 follows from the fact that $\frac{\sin(x)}{x} > \frac{2}{\pi}$ for $x \in \left[-\frac{\pi}{2}, \frac{\pi}{2}\right]$ and from Lemma 3, equation 40 follows from Lemma 4.*

Combining the results in Lemma 5 together with the ones in Lemma 1 we get the following result.

**Proposition 2** *For a spider web graph $G_{N,r}$ we have $\varphi_{N,r}(k) \leq \frac{2r}{2r-1}\sqrt{2\lambda_k^C}$, for $2 \leq k \leq N$.*

### E.1.2 RING PARTITIONS

Similarly as for wedge partitions, we will prove a bound on the ring Cheeger constants in terms of the eigenvalues of the path graph with $r$ vertices $P_r$. Some of the computations are analogous to the ones in the previous section, so we will skip the details for these.

We recall that the eigenvalues of $P_r$ are

$$\lambda_k^P = 1 - \cos\left(\frac{\pi(k-1)}{r-1}\right), \quad 1 \leq k \leq r, \tag{44}$$

see Chung (1997). We have the following inequality for the ring Cheeger constant.

**Lemma 6** $\sqrt{\lambda_k^P} \geq \frac{\sqrt{2}}{2}\frac{1}{2\lfloor\frac{r}{k}\rfloor}$, *for $2 \leq k \leq r$.*

**Proof 6** *It holds that*

$$\sqrt{\lambda_k^P} = \sqrt{1 - \cos\left(\frac{\pi(k-1)}{2(r-1)}\right)} \tag{45}$$

$$= \sqrt{2}\sin\left(\frac{\pi(k-1)}{2(r-1)}\right) \tag{46}$$

$$\geq \sqrt{2}\frac{2}{\pi}\left(\frac{\pi(k-1)}{2(r-1)}\right) \tag{47}$$

$$\geq \frac{\sqrt{2}}{2}\frac{k}{r-1} \tag{48}$$

$$= \frac{\sqrt{2}}{2}\frac{1}{\frac{r}{k} - \frac{1}{k}} \tag{49}$$

$$\geq \frac{\sqrt{2}}{2}\frac{1}{\lfloor\frac{r}{k}\rfloor + 1 - \frac{1}{k}} \tag{50}$$

$$\geq \frac{\sqrt{2}}{2}\frac{1}{2\lfloor\frac{r}{k}\rfloor} \tag{51}$$

$$\tag{52}$$

*where the inequality equation 51 follows from the fact that*

$$\lfloor\frac{r}{k}\rfloor + 1 - \frac{1}{k} \leq 2\lfloor\frac{r}{k}\rfloor. \tag{53}$$

Combining the results in Lemma 5 together with the ones in Lemma 6 we get the following result.

**Proposition 3** *For a spider web graph $G_{N,r}$ we have $\psi_{N,r}(k) \leq \sqrt{2\lambda_k^P}$, for $2 \leq k \leq N$.*

## F  PSEUDO CODES OF ALGORITHMS

In this section we give pseudo codes for algorithms, including Ring and Wedge Transformation, Valume and Cut calculation, WRT, PPO and full training pipeline.

We also provide the anonymized source code in the following link: https://anonymous.4open.science/r/K24-00F8/

---

**Algorithm 1:** Ring Transformation

---

**Input:** graph $G = (V, E, W, o)$
**Output:** Converted line graph $G_l$
$r \leftarrow$ radius of $V - o$;
**for** *each element $i$ from $1$ to $|r|$* **do**
    // rank of $r[i]$ in the sorted list of $r$
    $\text{Index}[i] \leftarrow \sum_{j=1}^{n} \mathbf{1}(r[j] \leq r[i])$;
**for** *each element $i$ from $1$ to $|E|$* **do**
    $E_{new}[i] \leftarrow \{\text{Index}[E[i].x], \text{Index}[E[i].y]\}$;
**for** *each element $i$ from $1$ to $|V|$* **do**
    $V_{new}[i] \leftarrow (\text{Index}[i], 0)$;
**return** $G_c = (V_{new}, E_{new}, W, (0,0))$

---

## G  DYNAMIC PROGRAMMING ALGORITHM FOR RING PARTITION PHASE

We show the pesudo-code of dynamic programming algorithm used in Ring Partition phase in Algorithm 9. This allows us performing ring partition only once. The time complexity of this algorithm is

**Algorithm 2:** Wedge Transformation

**Input:** graph $G = (V, E, W, o)$
**Output:** Converted circle graph $G_c$
$a \leftarrow$ angles of $V - o$;
**for** *each element $i$ from 1 to $|a|$* **do**
  // rank of $a[i]$ in the sorted list of $a$
  $\text{Index}[i] \leftarrow \sum_{j=1}^{n} \mathbf{1}(a[j] \leq a[i])$;
**for** *each element $i$ from 1 to $|a|$* **do**
  $a_{new}[i] \leftarrow \frac{2\pi}{|a|}\text{Index}[i]$ ;
**for** *each element $i$ from 1 to $|E|$* **do**
  $E_{new}[i] \leftarrow \{a_{new}[E[i].x], a_{new}[E[i].y]\}$;
**for** *each element $i$ from 1 to $|a_{new}|$* **do**
  $V_{new}[i] \leftarrow (\sin(a_{new}[i]), \cos(a_{new}[i]))$;
**return** $G_c = (V_{new}, E_{new}, W, (0, 0))$

**Algorithm 3:** Volume and Cut for Line

**Input:** Line graph $G_l = (V, E, W, o)$
**Output:** Cut $Cut$ and Volume $Volume$
$a \leftarrow$ angles of $V - o$;
**for** *$e, w$ in $E, W$* **do**
  **if** *$e.x < e.y$* **then**
    $x, y \leftarrow e.x, e.y$
  **else**
    $x, y \leftarrow e.y, e.x$
  **for** *$i$ from $x$ to $y$* **do**
    $Cut[i] \leftarrow Cut[i] + w$;
  **for** *$i$ from 1 to $x$* **do**
    **for** *$j$ from $y$ to $|V|$* **do**
      $Volume[i, j] \leftarrow Volume[i, j] + w$;
**return** $Cut, Volume$

**Algorithm 4:** Volume and Cut for Circle

**Input:** Circle graph $G_c = (V, E, W, o)$
**Output:** Cut $Cut$ and Volume $Volume$
$a \leftarrow$ angles of $V - o$;
**for** *$e, w$ in $E, W$* **do**
  $x, y \leftarrow e.x, e.y$ **if** *$e.x > e.y$* **then**
    $y \leftarrow y + |V|$;
  **for** *$i$ from $x$ to $y$* **do**
    $Cut[i \% |V|] \leftarrow Cut[i \% |V|] + w$;
  **for** *$i$ from 1 to $x$* **do**
    **for** *$j$ from $y$ to $|V|$* **do**
      $Volume[i, j] \leftarrow Volume[i, j] + w$;
**for** *$i$ from 1 to $|V|$* **do**
  **for** *$j$ from 1 to $i - 1$* **do**
    // when $i > j$, means the direction that cross n-to-1 part
    $Volume[i, j] = Volume[j, i]$;
**return** $Cut, Volume$

**Algorithm 5:** WRT Transformer with Ring Partition

**Input:** Line graph $G_l = (V, E, W, o)$, current partition $P$
**Output:** Embeddings for each nodes $emb$
$Cut, Volume \leftarrow Alg4(G_c)$;
// shape [N, 1] to [N, H] ;
$x \leftarrow Linear(Cut)$;
// shape [N, N, 1] to [N, N, H] to [N, N, 1] ;
$VMask \leftarrow Linear(Volume)$ ;
$PMask[i, j] \leftarrow 0$ if i and j are in same partition ;
$PMask[i, j] \leftarrow -\infty$ if i and j are in different partition ;
// Pos is positional embedding, in circle partition, just same as normal NLP Transformer
  $H_0 = x + \text{Pos}$ ;
// L is layer number ;
**for** *i from 1 to L* **do**
  $Q, K, V \leftarrow \text{Linear}(H_{i-1})$ ;
  $A \leftarrow QK^T + VMask + PMask$ ;
  $H_i' \leftarrow \text{Norm}(AV) + H_{i-1}$ ;
  $H_i \leftarrow \text{Norm}(\text{Linear}((H_i')) + H_i'$ ;
**return** $H_L$

**Algorithm 6:** WRT Transformer with Wedge Partition

**Input:** Circle graph $G_l = (V, E, W, o)$, current partition $P$
**Output:** Embeddings for each nodes $emb$
$Cut, Volume \leftarrow Alg3(G_c)$;
// shape [N, 1] to [N, H] ;
$x \leftarrow Linear(Cut)$;
// shape [N, N, 1] to [N, N, H] to [N, N, 1] ;
$VMask \leftarrow Linear(Volume)$ ;
$PMask[i, j] \leftarrow 0$ if i to j are in same partition ;
$PMask[i, j] \leftarrow -\infty$ if i and j are in different partition ;
// Pos is positional embedding, x-y coords on the circle $H_0 = x + \text{Pos}$ ;
// L is layer number ;
**for** *i from 1 to L* **do**
  $Q, K, V \leftarrow \text{Linear}(H_{i-1})$ ;
  $A \leftarrow QK^T + VMask + PMask$ ;
  $H_i' \leftarrow \text{Norm}(AV) + H_{i-1}$ ;
  $H_i \leftarrow \text{Norm}(\text{Linear}((H_i')) + H_i'$ ;
**return** $H_L$

**Algorithm 7:** PPO with Embeddings

**Input:** Embeddings for each nodes, i.e. actions, $emb$
**Output:** Action policy logits $a$, and critic for current state $v$
// [N, H] to [N, H] to [N, 1] ;
$a \leftarrow \text{Linear}(\text{Activate}(\text{Linear}(emb)))$ ;
// [N, H] to [1, H] to [1, 1] ;
$c \leftarrow \text{Linear}(\text{Activate}(\text{Attention}(emb)))$ ;
**return** $a, c$

**Algorithm 8:** Full Training Pipeline

**Input:** Graph $G = (V, E, W, o)$, target partition number $P_{max}$, target ring partition number $P_c$
**Output:** Next partition $a$
$P \leftarrow \{V\}\}$ ;
$samples \leftarrow \{\}$ ;
**while** *not converge* **do**
    // perform $P_{max}$ steps to generate partition and save into samples **for** *i from 1 to $P_{max}$* **do**
        **if** $|P| <= P_c$ **then**
            // do ring partition $G_l \leftarrow$ GraphToLine$(G)$ ;
            $Emb \leftarrow$ WRTWithRing$(G_l, P)$ ;
            $p, critic \leftarrow$ PPO$(Emb)$ ;
            $a \leftarrow$ sample action from $p$ ;
            $r \leftarrow$ radius of $G_l.V[action]$ ;
            $P' \leftarrow$ partition p by circle with radius $r$ ;
        **else**
            // do wedge partition $G_c \leftarrow$ GraphToCircle$(G)$ ;
            $Emb \leftarrow$ WRTWithWedge$(G_l, P)$ ;
            $p, critic \leftarrow$ PPO$(Emb)$ ;
            $a \leftarrow$ sample action from $p$ ;
            $angle \leftarrow$ angle of $G_l.V[action]$ ;
            $P' \leftarrow$ partition p by wedge with angle $angle$ ;
        **if** $|P| = P_{max}$ **then**
            $r \leftarrow$ NormalizedCut$(G, P)$
        **else**
            $r \leftarrow 0$
        $samples.add((G, P, p, critic, a, r))$ ;
        $P \leftarrow P'$
    // calculate loss and train with samples ;
    **if** $|samples| = target\_size$ **then**
        **for** $sample$ in $samples$ **do**
            $p_{old}, c_{old}, r \leftarrow sample$ // here use sample as PPO input, in fact sample will do same
              as above to calculate p and critic. $p, critic, critic' \leftarrow$ PPO$(sample)$ ;
            $adv \leftarrow r - \gamma critic' + critic$ ;
            $loss_p \leftarrow$ clip$(p/p_{old} * adv)$ ;
            $loss_v \leftarrow (r - \gamma critic' + critic)^2$ ;
            $loss_{ent} \leftarrow$ Entropy$(p)$ ;
            $L \leftarrow w_p loss_p + w_v loss_v + w_{ent} loss_{ent}$ ;
            Backward loss $L$ ;
        $samples \leftarrow \{\}$

$O(n^2 k)$. The DP matrix $dp[i, j]$ stores the minimum normalized cut value when partitioning the first $i$ nodes into $j$ segments, with transitions recorded in the predecessor matrix $pre[i][j]$. The final loop traces back from the last segment's optimal value to reconstruct the partition indices by following $pre$ entries iteratively.

---

**Algorithm 9:** Dynamic Programming for Ring Partition

---

**Input:** Precomputed cut weight matrix $Cut$, volume matrix $Volume$, number of partitions $k$
**Output:** Optimal Normalized Cut $res$, partition indices $P$
$sector\_nc[i, j] \leftarrow (Cut[i] + Cut[j])/Volume[i, j]$ for all $i, j$;
// $dp[i, j]$ means the best result when we perform partition on node $i$ and it is the $j$-th partition
$dp[i, j] \leftarrow \infty$ for all $i, j$;
$dp[0, 0] \leftarrow 0$;
// $pre[i, j]$ records where the value for $dp[i, j]$ transits from
$pre[i, j] \leftarrow 0$;
**for** $i$ *from* 1 *to* $|Cut| - 1$ **do**
    **for** $j$ *from* 1 *to* $k - 1$ **do**
        // enumerate all $p < i$ and assume last partition is from $p$ to $i$
        **for** $p$ *from* 1 *to* $i - 1$ **do**
            $agg\_res[p] \leftarrow \max(dp[p, j - 1], sector\_nc[p, i])$;
        $pre[i, j] \leftarrow \arg\min(agg\_res)$;
        $dp[i, j] \leftarrow agg\_res[argmin]$;
// The last partition should be from $p$ to $|Cut|$, update it to $dp[p, k - 1]$
**for** $p$ *from* 1 *to* $|Cut| - 1$ **do**
    $dp[p, k - 1] = \max(dp[p, k - 1], sector\_nc[p, |Cut| - 1])$;
$result \leftarrow dp[res\_x, res\_y]$;
// get final partition indices
$r_x \leftarrow \arg\min(dp[:, k - 1])$;
$r_y \leftarrow k - 1$;
$P \leftarrow \{\}$;
**while** $r_y > 0$ **do**
    $P \leftarrow P \cup \{r_x\}$;
    $r_x \leftarrow pre[r_x, r_y]$;
    $r_y \leftarrow r_y - 1$;
**return** $result, P$

---

