# OpenReview forum: "Solving Normalized Cut Problem with Constrained Action Space"
_ICLR.cc/2025/Conference — Submitted to ICLR 2025_

### Official Review · Reviewer_Adwf · 2024-10-31

**Soundness:** 3
**Presentation:** 3
**Contribution:** 3
**Rating:** 6
**Confidence:** 3

**Summary:**

The manuscript presents Wedge and Ring Transformers (WRT), an RL-based method for solving the Normalized Cut (NC) problem in weighted graphs with shape-specific constraints. By transforming graphs into polar coordinates and using Transformers with Proximal Policy Optimization, WRT effectively handles both ring and wedge partition shapes, optimizing NC while adhering to these constraints.

**Strengths:**

1. The paper addresses the Normalized Cut problem in the context of real-world applications, such as road network simulations, where partition shape constraints are critical.
2. The introduction of the Wedge-Ring Transformer, tailored to handle specific shape constraints in graph partitioning, is innovative.

**Weaknesses:**

1. The paper includes a limited set of baseline methods for comparison. Adding more baselines, particularly those used in NeuroCUT, would strengthen the evaluation by providing a more comprehensive assessment of WRT's performance.
2. The baselines lack specialized adaptations for the "Ringness" and "Wedgeness" constraints, while WRT is explicitly designed with these constraints in mind. This discrepancy may lead to an unfair comparison, as the baselines are not optimized to meet these specific structural requirements.
3. The experiments use relatively small graph instances, whereas NeuroCUT and other methods operate on benchmarks with thousands of nodes, aligning more closely with real-world scales. The current experimental scale may limit the ability to assess WRT’s applicability to large-scale, practical scenarios.
4. Given the use of Transformers, I am concerned about the performance and computational cost of training and inference on large-scale datasets.

**Questions:**

1. Were NeuroCUT and ClusterNet evaluated by training on the same datasets as WRT? Ensuring consistent training conditions is crucial for fair comparison.
2. The Cheeger Bound presented appears to be a specific case of a more general result. How does this theoretical finding contribute to model design or provide insights for experimental evaluation?
3. What specific metrics are used in Tables 1 and 2?
4. Why are there no generalization results for NeuroCUT and ClusterNet in Table 2?
5. According to Fig. 6, it appears that RL training converges early (10–20k of 400k steps). Does the extended training beyond this point contribute to any performance improvements, or could training resources be optimized?

---

> ### Author Response · Authors · 2024-11-23
>
> We truly appreciate the points you raised in your review, as they help us improve and refine our work.
> Below, we’ve outlined our responses to address the issues:
>
> Weakness Part
> ---
>
> **Including more baselines**
>
> Thank you for the suggestion. From the results of NeuroCUT and ClusterNet, they outperform baselines in all experiments, so we chose them as the comparism methods and omit the weaker competitors.
>
> **Introducing Ringness and Wedgeness in compared methods**
>
> Our ultimate goal in this problem is to provide a superior parition with smaller Normalized Cut, and without extra constraints.
> Baselines cannot incorporate the Ringness and Wedgeness constraint in their structure, because they directly do partition based on nodes, or from a existing partition (e.g. the partition result given by METIS).
> Without the constraint, they should have a larger action space compared with WRT; however, they actually cannot perform better results.
> On the other hand, although our method restricts the action space, we finally find partition with smaller Normalized Cut.
> This further demonstrates that introducing the domain knowledge by constraining the action space represents a more effective approach.
>
> **Graphs are relatively small**
>
> Thank you for your question. During training, we need to randomly sample a sub-graph during every iteration and perform checks such as connectivity and map coverage on that graph. Currently, this sampling process is inefficient, and as the node number increases, the overhead rises significantly. This has temporarily prevented us from conducting larger-scale training. We are also working on improving processing efficiency so that the model can be applied to maps with a larger number of points.
>
> **Performance of Transformers in large graphs**
>
> In this paper, our primary contribution lies in presenting a novel approach to incorporating domain knowledge by constraining the action space. Specifically, we restrict the action space to ring and wedge, providing better results with Transformers.
> Although Transformers is able to perform good results when scaling to larger graphs, the quadratic time complexity of Transformers indeed poses challenges.
> To address this issue, we can employ acceleration techniques such as Flash Attention or Linear Attention to alleviate the problem.

---

> ### Author Response · Authors · 2024-11-23
>
> Questions part
> ---
>
> **Training settings for NeuroCUT and ClusterNet**
>
> Yes, all learning based methods used the same training datasets and test datasets while maintaining the same number of training steps. It is important to note that these two algorithms are designed for unweighted graphs and do not directly support weighted graphs. Therefore, we made the following modifications:
>
> - For ClusterNet, we replaced the values of the 0-1 adjacency matrix with edge weights. Additionally, we adjusted its loss calculation formula to align with weighted Normalized Cut.
>
> - For NeuroCUT, we also added the support of weighted edges. Specifically, we adjusted the reward function to match the weighted Normalized Cut and incorporated edge weight information into the node embeddings, including the weights of adjacent edges and embeddings obtained through random walks. Furthermore, NeuroCUT requires an initial partition, and its built-in initial partition method has poor performance; thus, we replaced it with the results from METIS.
>
> **Contribution of Cheeger bounds**
>
> The Cheeger bound presented in this work is not a specific result of a more general case. Note that, in our work the Cheeger constant corresponds to the minimum normalized cut. For classical Cheeger bounds, the Cheeger constant is found by minimizing the normalized cut over all possible partitions. In our case, we don't consider the class of all possible partitions, but only on the restricted class of ring+wedge partitions. So, the minimum "constrained" normalized cut is be greater or equal than the classical Cheeger constant(=minimum normalized cut). In principle it is not know how large this quantity is and if it still makes sense to minimize the normalized cut on this smaller space of partitions. In our work we prove that actually also the $k$-th "constrained" normalized cut is still bounded from above by $\mathcal{O}(\sqrt{\lambda_k})$, following a behavior similar to the classical case. This shows that constraining the minimization to this class of partitions leads to good quality normalized cuts. Proving this result in full generality is very difficult, but showing this already for spider-web graphs provides a good justification of using ring+wedge partitions. Thank you for your question, we will clarify this point in the final version of the paper.
>
> **Metrics of Table 1 and 2**
>
> The metrics of Table 1 and 2 are Normalized Cut of partitions provided by different methods, which is mentioned in the caption of the Tables. A lower value indicates better performance.
>
> **Generalization results for compared methods**
>
> NeuroCUT and ClusterNet does not support generalization directly based on their codes. We are working on modifying them to support generalization, and give their generalization results in the revised version.
>
> **Training curves of RL**
>
> Although the reward seems converge early, there is still slight optimization for the final performance.
> When wedge partition is not fixed, there is rapid initial convergence, but continued training leads to a decline in the reward before reaching the best solution, resulting in the selection of the best checkpoint being from the early stages.
> In contrast, fixing wedge partition allows for ongoing exploration around a favorable position, yielding better results.
>
> We have also added reward curves during the testing in Figure 6 in the Appendix B.4, from the curves we can find the performance is increasing steadily with fixed wedge partition, and when wedge partition is not fixed, it cannot have further performance increase after initial converge.
>
> Regarding the number of training steps, this is a useful suggestion.
> Indeed, the optimization achieved in the later stages of our current training strategy is significantly smaller compared to the beginning. We will explore better training strategies to accelerate the convergence process during this phase.

---

> > ### Comment · Reviewer_Adwf · 2024-11-24
> > **To authors**
> >
> > Thank you to the authors for their response. I still have the following concerns:
> > 1. Given the completely different data and scenarios, I strongly recommend that the authors include baselines from NeuroCUT and ClusterNet in subsequent versions of the paper.
> > 2. I agree with the approach of incorporating expert knowledge into the model, as it can help address specific problems. However, I believe there may be an inherent unfairness in comparing the specified method with general-purpose methods. In a certain sense, this comparison may even be framed as addressing a different problem. Perhaps incorporating expert knowledge into those general-purpose methods could also yield promising results. I believe this is an adaptation worth exploring and should be included in the comparison to highlight the effectiveness of the proposed method.
> > 3. The motivation of the paper is to solve the normalized cut for traffic data, however, the proposed method currently seems difficult to apply to such real-world scenarios. This discrepancy is my main concern at the moment. As you have mentioned, I suggest that the authors consider larger-scale instances in future versions and explore the use of advanced transformer models to alleviate the computational complexity.
> >
> > Given the above, I will maintain my current score.

---

> > > ### Author Response · Authors · 2024-11-29
> > >
> > > Thank you for your response. Regarding your concerns, we provide the following explanations:
> > >
> > > 1. We have included additional comparison methods from NeuroCUT, specifically DMon, MinCutPool, and Ortho, presenting their performance in section B.7 of the Appendix. We conducted hyper-parameter searches to determine the best parameters for these methods. The results indicate that their performance is significantly inferior to that of WRT.
> > > 2. Thank you for your suggestion. In fact, we also attempted similar approaches but found that general-purpose methods struggle to incorporate ring and wedge constraints, which is a primary reason for the development of WRT. Existing methods either adjust nodes based on a pre-existing partition or directly provide different class probabilities for each point. The former relies on an existing partition and makes it challenging to maintain the partition as a ring and wedge partition when only adjusting single nodes. The latter similarly faces difficulties in constraining the overall shape during probability generation. Therefore, we believe that the WRT method is an effective attempt to appropriately apply domain knowledge to constrain the action space, coupled with a corresponding method for solution finding.
> > > 3. Thank you for your suggestion. We will consider larger-scale datasets in future work and explore the utilization of more advanced transformer models to alleviate computational complexity.
> > >
> > > Once again, thank you for your response, and we hope these answers can address your concerns.

---

> > > > ### Comment · Reviewer_Adwf · 2024-12-03
> > > >
> > > > Thank the authors for the reply. This paper indeed presents an interesting and novel idea, so I would like to maintain the rating. But considering the current shape of the paper, I cannot fight for an acceptance.

---

### Official Review · Reviewer_rJiE · 2024-11-03

**Soundness:** 3
**Presentation:** 3
**Contribution:** 2
**Rating:** 5
**Confidence:** 3

**Summary:**

This paper proposes the Wedge Ring Transformer (WRT), an RL-based approach to minimize the Normalized Cut (NC) on planar weighted graphs. WRT leverages polar coordinates and employs a multi-head transformer with a Proximal Policy Optimization (PPO) objective to address the NC problem. The approach utilizes a two-stage training process to effectively learn both ring and wedge partitioning strategies. Experimental results indicate that WRT effectively reduces the NC.

**Strengths:**

The paper provides a clear definition of the Normalized Cut (NC) problem and the description of the Wedge Ring Transformer (WRT) is well-articulated.
The design of transformations specifically tailored for ring and wedge shapes appears effective.
Provide some theoretical analysis about cheeger bound for ring and wedge partition.

**Weaknesses:**

Ablation studies: The ablation studies primarily focus on the two-stage training process, but lack analysis on key components of the paper's main contribution, such as the wedge-ring transformer, PAMHA, and pre-calculation. Ablation studies on these components would provide a more comprehensive evaluation of the WRT architecture.
Running Times:  The paper does not provide an analysis of the model's runtime, leaving the computational efficiency of WRT unaddressed.

**Questions:**

1.	The paper argues that GNNs were not used due to scalability issues. However, the proposed method also seems to require processing the entire graph at once, and experiments were conducted on data with a maximum of only 200 nodes. It remains unclear how WRT scales to larger graphs, and additional evidence of scalability would strengthen the paper's claims.
2.	What are the evaluation metrics in Table 1 and Table 2?
3.	Although WRT is designed for ring and wedge-shaped partitions, I am interested in understanding its performance on other types of datasets. For example, how does it perform on datasets that primarily feature extended ring shapes, such as the long-tail structures often found in knowledge graphs?

---

> ### Author Response · Authors · 2024-11-23
>
> We truly appreciate the points you raised in your review, as they help us improve and refine our work.
> Below, we’ve outlined our responses to address the issues:
>
> **More ablation studies**
>
> Thank you for pointing out the problem.
> We are working on implementing ablation studies you mentioned, and will add them in the revised version of the paper later.
>
> **Running time of our method**
>
> | Method      | 50           | 100          | 200          |
> |-------------|--------------|--------------|--------------|
> | METIS       | 0.073082209  | 0.06215477   | 0.077347279  |
> | Spectral    | 0.010553621  | 0.034367323  | 0.506824732  |
> | NeuroCUT    | 0.844880027  | 1.596169194  | 2.511722407  |
> | ClusterNet  | 0.140056022  | 0.145348837  | 0.140056022  |
> | WRT         | 0.049610889  | 0.072737801  | 0.26129086   |
>
> We give the inference time of our method and other competitors on City Fraffic graphs with 4 partitions in the table above.
> From the table, We can observe that METIS and ClusterNet have relatively low and stable running time; Spectral Clustering, while also having a shorter running time in the experiments, exhibits a rapid increase based on node number. NeuroCUT, takes a longer time and shows a significant growth as the number of points increases. Our method WRT takes relatively longer than METIS and ClusterNet, but is significantly faster than NeuroCUT.
>
> **Comparison between GNNs and Transformers**
>
> In this work, we choose Transformers instead of GNNs due to their superior scalability for larger graphs. In Transformer, each token can attend to all other tokens, enabling information exchange to occur in parallel. Conversely, GNNs are limited to observing only neighboring nodes and typically require a significantly greater number of layers to capture global information.
>
> The Normalized Cut, particularly when applied with ring and wedge, necessitates a holistic view of the global graph. Since GNNs inherently lack access to this global information, this limitation represents a significant bottleneck for their ability to learn effective strategies for Normalized Cut.
>
> **Graphs are relatively small**
>
> Thank you for your question. During training, we need to randomly sample a sub-graph during every iteration and perform checks such as connectivity and map coverage on that graph. Currently, this sampling process is inefficient, and as the node number increases, the overhead rises significantly. This has temporarily prevented us from conducting larger-scale training. We are also working on improving processing efficiency so that the model can be applied to maps with a larger number of points.
>
> **Metrics of Table 1 and 2**
>
> The metrics of Table 1 and 2 are Normalized Cut of partitions provided by different methods, which is mentioned in the caption of the Tables. A lower value indicates better performance.
>
> **Performance on other types of datasets**
>
> Our core idea is to constrain the action space based on domain knowledge, thereby training a better model.
> In this paper, Our method focuses on the road network graph and proposes the ring-wedge partition method.
> However, for knowledge graphs, since the nodes are not positioned on a plane, directly applying ring and wedge shape may be challenging. One idea could be proposing a mapping algorithm to project the nodes onto a plane for analysis. Alternatively, we could explore new methods to simplify the action space based on the features of the knowledge graphs.

---

> > ### Comment · Reviewer_rJiE · 2024-11-26
> > **To authors**
> >
> > Thank the authors for the response. They have only partially addressed my concerns. I will keep the score unchanged at this time.

---

> > > ### Author Response · Authors · 2024-11-29
> > >
> > > Thank you for your response. In the latest revised paper, we made the following changes:
> > >
> > > 1. For the comparism between GNNs and Transformers, we conducted a comparison between WRT, GNN and vanilla Transformers in section B.6. We selected GPSGNN, which integrates both local interactions with GNN and global interactions with Transformers. We also applied Transformers directly to transformed graphs without Pre-Calculation and PAMHA. The results indicate that GNN fails to provide reasonable partition results, even with the assistance of Transformer for global interaction. In contrast, when graph transformation is applied, the performance improves significantly, suggesting that transforming the graph into a sequential format greatly enhances the partitioning task. Finally, the application of WRT leads to further substantial improvements.
> > > 2. We conducted additional ablation studies in Section B.6 to validate the efficacy of Pre-Calculation and PAMHA. The difference between WRT and conventional Transformers is the implementation of Pre-Calculation and PAMHA; since PAMHA is related to Pre-Calculation, disabling Pre-Calculation effectively means disabling WRT. The results illustrate that following graph transformation, we observe significant enhancements. We also find that both Pre-Calculation and PAMHA contribute positively to the overall performance of WRT.
> > >
> > > We hope these modifications address more of your concerns.

---

### Official Review · Reviewer_YY7v · 2024-11-04

**Soundness:** 2
**Presentation:** 2
**Contribution:** 2
**Rating:** 3
**Confidence:** 4

**Summary:**

This work tackles a special case of a normalized-cut problem: that of spider-web shaped weighted planar graphs.
The graph is partitioned into rings, and the outer ring is partitioned into wedges. The approach transforms the graph by:
a. projecting ring nodes onto an axis according to their distance from a center while maintaining node order
or by
b. projecting nodes onto a unit circle.
The transformation results in the partitioned nodes forming a sequence, which is encoded by a transformer.
Reinforcement learning is used to find the ring radius and number of outer ring wedges that result in a minimal normalized cut.

Specifically, PPO is used, where the state, action, and rewards are encoded as:
a. State is the graph, number of rings and wedges of the outer ring.
b. Actions are the ring radius or wedge angle.
c. Rewards are 0 in all steps, and the negative normalized cut at the end.
The wedge partition is trained using random ring partitions, followed by training of both ring and wedge partitions.
The ring partition is first inferred during testing, followed by the wedge partition.

This work demonstrates that this transformation is suitable for a specific case of road networks.
The transformation is applied as a preprocessing step, finding a minimal normalized cut with a lower value than other baselines.

The approach is evaluated using synthetic and real-world data.
a. 400k spider-web shape synthetic graphs with a 50 or 100 nodes, ring and wedge partitions, with unweighted and random edge weights.
b. Connected sub-graphs randomly extracted from real-world city maps with edge weight corresponding to traffic.

The performance of the approach is compared with a baseline partitioning method, METIS, and with spectral clustering.
The ring and wedge partitions are compared with brute force and random partitions.

**Strengths:**

1. The graph transformation is applied as a pre-processing step, aiming to utilize the specific graph structure.

2. The results are a minimal normalized cut with a lower value than other trivial baselines.

**Weaknesses:**

1. The decisions to apply the transformations to the graph are manual.
The method and its implementation details are ad-hoc and very specific.

2. Dynamic programming is used to compute the optimal partition given the maximum radius and ring count. Ablation studies of this algorithm and the reinforcement learning approach are missing.

3. The graphs are relatively small consisting of 50, 100 (for training), or 200 nodes (in testing).

**Questions:**

Can this approach be automated by classifying the graphs to automatically find which transformations should be applied as a preprocessing step?

---

> ### Author Response · Authors · 2024-11-23
>
> We truly appreciate the points you raised in your review, as they help us improve and refine our work.
> Below, we’ve outlined our responses to address the issues:
>
> **Method is ad-hoc and specific**
>
> The graph transformations are not purely manually designed; the transformations we propose are based on the application scenario of planar road networks. We are inspired by real-world road network designs. Theoretical analysis also supported the notion that ring and wedge partition form a robust constrained action space. We believe that this approach can be applied to various road network-related problems.
>
> **Comparison between Dynamic Programming and Reinforcement Learning**
>
> Dynamic programming is a part of our method. Once the radius and the number of rings are determined, we can use dynamic programming to find the optimal solution for the ring partition; meanwhile, RL is employed to seek the optimal partition for the wedge part. Combining both parts completes the ring-wedge partition and their performance cannot be directly compared.
>
> **Graphs are relatively small**
>
> Thank you for your question. During training, we need to randomly sample a sub-graph during every iteration and perform checks such as connectivity and map coverage on that graph. Currently, this sampling process is inefficient, and as the node number increases, the overhead rises significantly. This has temporarily prevented us from conducting larger-scale training. We are also working on improving processing efficiency so that the model can be applied to maps with a larger number of points.
>
> **Can this approach be automated by classifying the graphs to automatically find which transformations should be applied as a preprocessing step?**
>
> Thank you for the suggestion. Our method focuses on the road network graph and proposes the ring-wedge partition method based on the characteristics of the road network graph, which enhances the performance of reinforcement learning (RL) by constraining the action space. For other graphs, we can also design similar constraints on action space based on the inherent properties of the graph, thereby training better RL models. We believe that constraint design is closely related to specific problems and requires a solid theoretical foundation, thus necessitating tailored one-on-one designs. Automatically determining the appropriate transformations as a pre-processing step is currently out of scope. In fact that is a completely different research problem.

---

> ### Comment · Reviewer_YY7v · 2024-11-27
> **To authors**
>
> Thank you authors for your response.
> Perhaps an "apples to apples" comparison with other methods and on larger instances would improve the work.
> The current approach is very specific and ad-hoc and demonstrated on relatively small instances.
> I will keep my review score unchanged.

---

> > ### Author Response · Authors · 2024-11-29
> >
> > Thank you for your valuable feedback. We will enhance the generalizability of this method to graphs with more instances in our future work and provide experimental results on larger graphs.

---

### Official Review · Reviewer_8zQE · 2024-11-08

**Soundness:** 3
**Presentation:** 2
**Contribution:** 2
**Rating:** 5
**Confidence:** 3

**Summary:**

The paper describes a Reinforcement Learning strategy to solve an approximate minimum normalized cut on spider web-like planar graphs, like city street maps.  The idea is that the problem can be approximated by a circles-wedges clusterings, in which inner nodes (w.r.t. some central point o) can be grouped w.r.t. their distance from a center o, while outer nodes are further subdivided w.r.t. their angular polar coordinates. The actions to be performed will then be the radius of the outer circle and the (discrete) points where to split the outer nodes.
Some training strategies are defined to help the problem converge and refine the grouping.
The method is tested on synthetic spider web-like graphs, and subgraphs extracted from a city map.

**Strengths:**

I like the idea of modeling the grouping of nodes according to the previous knowledge about the domain. This allows to simplify the minimum graph cut problem for the specific type of graphs considered, and obtain better results than other general-purpose grouping algorithms.

**Weaknesses:**

My main concern is about the quite demanding assumption of the algorithm, which is designed to work on spider-like planar graphs, where nodes are embedded (have coordinates) in R^2. In particular, my comments are:
- even if the proposed solution is sound for the specific problem, I’m not sure it is general enough to be of broad interest to this community. It looks more suited for a venue in the specific application.
- it is not explained why the grouping in inner circles and outer wedges is a good modeling. Is it a pattern observable in other city map grouping algorithms? Does this pattern apply to all cities?

There is a drop in writing quality in section 5, which raised some doubts:
- It is incorrect to say that transformers work only on sequences, they work on any set of points but often benefit from a positional encoding.
- Sections 5.2.1 and 5.2.2 are quite intricate and could be simplified. In practice, they define two different positional encodings for ings and wedges, where points for the ring are encoded with their distance from the center, and for wedges, they are projected into the unit circle (and possibly equispaced?).
- The optimal partitioning of circles (row 322) should be better introduced.
- In 5.4 I don’t understand what the “Current Partition” is. It is represented by a binary mask? How is it converted into the colored square matrix in Figure 4?

To broaden the impact of the work, it would be worth trying to apply the proposed method to different families of graphs and different datasets. Also, graph cut methods seem to exist specifically designed for planar graphs (e.g. “Efficient Planar Graph Cuts with Applications in Computer Vision”) that would be worth considering in the comparison.

**Questions:**

- Your setting is much simpler than finding normalized min cut in general undirected graphs. Is it still a NP-Hard problem? For instance, polynomial algorithms for the min-cut on planar graphs exist (I just found a few, but I might be missing some fundamental details). Would they also apply to your definition of normalized cut?
- From reading the text, it sounds like you are providing a novel definition of normalized cut, but it looks like the standard definition to me. Am I missing something? My confusion is further increased by the statement at line 214: “Despite being a simpler class of graphs, these bounds give a theoretical justification of the normalized cut definition equation 2 and the ring-wedge shaped partition.”
- At row 283 you write “Note that this transformation does not change the order of the nodes or the partitions.” What do you mean by node order?
- How is the center of the graph defined?

---

> ### Author Response · Authors · 2024-11-23
>
> We truly appreciate the points you raised in your review, as they help us improve and refine our work.
> Below, we’ve outlined our responses to address the issues:
>
> Weakness Part
> ---
>
> **Generalizability of our method**
>
> Our method is applicable to a broad subset of planar graphs, particularly those modeling transportation and road networks, like traffic simulation, map representation learning, and traffic prediction. This makes our method suitable for a broad subclass within spatio-temporal application domains. Furthermore our method provides a principled ways of incorporating prior knowledge into partitioning using transformer-based reinforcement learning.
>
> **Modeling in ring-wedge style**
>
> Many city road networks naturally exhibit a ring-wedge structure, where traffic flows from outer suburban or remote areas (wedge-shaped regions) into the city center, which is often organized into multiple concentric rings. For instance:
>
> 1. Beijing's Five Ring includes a large circular expressway encircling the city, acting as the outermost "ring" within the urban area, while traffic flows from various surrounding suburban areas converge toward the city center.
> 2. Shanghai's urban expressway system consists of three ring roadways and two major cross roads in the central urban area, with traffic streams from the outer districts feeding into these rings.
> 3. Qatar's infrastructure also features multiple ring roads, designed to handle traffic coming from different directions and converging toward central hubs.
>
> Also, at the end of large events, people tend to expand outward from the center of the stadium, exhibiting a radial dispersion through the main avenues.
>
> This ring-wedge traffic pattern is prevalent in many urban networks, especially for large cities, making our method broadly applicable.
> By explicitly constraining the shape of network components to reflect these patterns, we address practical challenges and provide a principled approach to fine-grained, shape-controlled graph partitioning.
>
> On the theoretical front we provide new
> Cheeger inequalities that connect the spectral properties of a graph with
> algebraic properties that capture the shape of the partitions.
>
> **Transformers with Graphs**
>
> Thank you for pointing out the issue that our expression in the paper was not sufficiently precise. We intend to clarify that sequential input helps to simplify the input space, making it more suitable for Transformers' learning. We are currently revising the relevant content and will update the expression in the revised version.
>
> For the use of positional encoding, we are also implementing comparative experiments by not performing ring or wedge transformations, but directly use the coordinates as positional encoding and treat the edge weights as attention masks. We will update the results of the comparative experiments later.
>
> **Writing improvements in Section 5.2**
>
> Thanks for your kind suggestion. We are improving our writings on Section 5.2.1 and 5.2.2 to simplify the explanation according to your advice and will prepare the revised version.
>
> **Optimal Partitioning of Circles in line 322**
>
> Thanks for your advice. We have added the pseudo-code of finding Optimal Partitioning of Circles in the appendix G.
> The algorithm uses dynamic programming, which have time complexity $O(n^2k)$.
>
> **Current Partition, Binary Mask and colored Square matrix in Sec. 5.4**
>
> The *Current Partition* represents the indices of points on the selected wedge and indicates where we are splitting between wedge$_i$ and wedge$_{i+1}$ . It is a binary mask. Suppose there are $k$ wedges for the *Current Partition*. If we decide to split between $j$th wedge and $j+1$th wedge, it will only be related to the existing partitions that cover $j$. Therefore, in the attention mask, we mask out the portions associated with other partitions. We are working on providing more detailed and intuitive explanations in the appendix.
>
> **Applying on different datasets and relationship to graph cut methods**
>
> We appreciate your suggestion and have conducted a thorough review of the relevant literature. The methods employed in the referenced articles [1] apply to the min-cut problem. Min-cut and max-flow are dual problems. For general graphs, the time complexity of algorithms solving max-flow such as Dinic is $O(V^2E)$, while it can be reduced to $O(E \log E)$ for planar graphs.
> However, Normalized Cut differs from these approaches, as it requires simultaneous consideration of both the size of the cut and the weights of internal edges, making it unsuitable for resolution through min-cut methods. To the best of our knowledge, there is currently no polynomial-time solution or verification method available for Normalized Cut, even on planar graphs.
>
> [1] SCHMIDT, Frank R.; TOPPE, Eno; CREMERS, Daniel. Efficient planar graph cuts with applications in computer vision. In: 2009 IEEE conference on computer vision and pattern recognition. IEEE, 2009. p. 351-356.

---

> ### Author Response · Authors · 2024-11-23
>
> Questions Part
> ---
>
> **Is it still a NP-Hard problem**
>
> Thank you very much for bring up this point which we should have mentioned in the paper. The problem of finding the Normalized Cut (normalized min-cut) on planar graphs is classified as NP-Complete and may also be NP-Hard. As referenced in the appendix of [2], it has been established that determining the Normalized Cut on regular grids is at least NP-Complete due to a reduction from the PARTITION problem.
>
> [2] Shi, J., \& Malik, J. (2000). Normalized cuts and image segmentation. IEEE Transactions on pattern analysis and machine intelligence, 22(8), 888-905.
>
> **Definition of Normalized Cut and line 214**
>
> We followed the standard definition of Normalized Cut without introducing a new definition for it. When minimizing Normalized Cut, the partition can consist of arbitrary point sets. However, we found that the model struggles to learn a good partitioning strategy in an unconstrained setting. Based on prior knowledge of the traffic environment, we constrain the action space to *ring and wedge*, enabling the model to propose better strategies within a constrained action space. In Line 214, we consider a special case of planar graphs, known as *spider web graphs*. Under this specific case, we prove that *ring and wedge* partitions achieve the same bound as the classical case. This theoretically demonstrates the feasibility of restricting the action space to *ring and wedge*.
>
> **Node order in line 283**
>
> Thank you for pointing out the error in our expression. In fact, the first transformation, which maps the points to the x-axis, does not involve the node order. The node order only comes into play during the second step when adjusting the positions of the points on the coordinate axis. In the second transformation, we can move the points on the x-axis to positions $(X, 0)$ without changing their relative order, where $X$ is the radius order of the node among all nodes. Clearly, for any \textit{ring partition} on the transformed graph, we can find an equivalent \textit{ring partition} on the original graph.
>
> **Definition of the graph center**
>
> Thanks for pointing out the question.
> Currently, we directly use the average of all coordinates as the graph center. In practical applications, we can also select appropriate centers based on real-world situations, such as designating high-traffic locations like stadiums and concert venues as centers.
>
> We have added an ablation study in appendix B.5 to explore the the impact of selecting different centroids.
> We offset the centroid by a distance of up to 5\% and recalculated the results of Normalized Cut. A normalized value closer to zero indicates better performance.
> We observe that any offset from the centroid results in a worse performance, and with greater offsets correlating to a more significant decline.
> We also find that in nearly half of the cases where offsets were applied, the resulting errors remained within 5\%. Thus, in this paper, we opted to use the centroid as the center of the graph.
>
> Histogram also shows that in approximately 15\% of cases, offsetting the centroid yielded improvements of over 10\%.
> In the future, we can propose a more effective strategy for centroid selection to enhance the algorithm's performance.

---

> > ### Comment · Reviewer_8zQE · 2024-11-25
> > **Response to authors' rebuttal**
> >
> > I thank the reviewers for clarifying some of my doubts.
> > I understand that some big cities exhibit a ring-and-wedges structure, but I agree with other reviewers that this seems like an ad-hoc solution for a normalized cut algorithm. For instance, the assumption of perfectly concentric rings might be limiting. Including some degree of adaptation, such as some metric learning approach on points coordinates, could make the proposed approach more general.
> > Anyway, I will consider raising my score after the discussion phase with the other reviewers.

---

> > > ### Author Response · Authors · 2024-11-29
> > >
> > > Thank you for your reply. Although we introduced constraints of ring and wedge during partitioning, we optimize and adjust the partition with post-refinement, allowing the model to output non-pure ring and wedge partitions, which is shown in the Figure 4c(4). We have also validated the effectiveness of post-refinement in our ablation studies. Your feedback is very valuable, and we will explore more flexible partitioning methods to enhance our results in the future.
> > >
> > > Additionally, in our latest revised version, we included an ablation experiment that modifies only the positional encoding of Transformer. In Appendix B.6, the results with name Transformer show the performance of adjusting only the positional encoding, and without performing Pre-Calculation and PAMHA. The results indicate a significant improvement compared to direct input of the original graph; however, there is still a noticeable gap compared to WRT.
> > >
> > > Thank you once again for your valuable feedback!

---

### Author Response · Authors · 2024-11-29

Dear Reviewers and Area Chair,

We sincerely appreciate the valuable suggestions from reviewers. In our paper, we investigate the Normalized Cut problem on weighted planar graphs, noting that existing methods have failed to resolve this problem effectively. Inspired by the ring and wedge structures prevalent in urban environments, we constrain the action space of the partitioning process to ring and wedge and proved that, under certain constraints, the resulting partition exhibits the same upper bound in terms of the Cheeger Bound as general partitions. Based on this framework, we introduce the concepts of ring transformation and wedge transformation, along with the WRT model, achieving state-of-the-art results for this problem.

In response to the reviewers' concerns, we have implemented substantial improvements to the manuscript, including better expressions, additional comparative methodologies, more ablation studies, the investigation of graph centroid selection, detailed implementations of the dynamic programming algorithm in ring partitioning, and clearer performance curves during training. These enhancements significantly improve the completeness and persuasiveness of our paper.

The reviewers also pointed out the limitations regarding the graph size and the somewhat ad-hoc nature of our methods. We would like to emphasize that our primary contribution lies in presenting a novel approach for using reinforcement learning to tackle challenging problems by constraining the action space based on domain knowledge. Although in this paper we mainly explored Normalized Cut problem, this approach is broadly applicable, allowing for the design of constraints for other challenges through their domain knowledges. In future work, we plan to explore applying this method to larger-scale graphs and a wider range of graph partitioning problems, as suggested by the reviewers.

Once again, we extend our gratitude to the reviewers for their invaluable feedback and to the area chair for their support.

Sincerely,
The Authors

---

### Meta-Review · Area_Chair_m94Z · 2024-12-18

**Metareview:**

This paper introduces Wedge and Ring Transformers (WRT), the first method to explicitly constrain Normalized Cut (NC) partitions to specific shapes like rings and wedges on planar graphs. Leveraging polar coordinates and a transformer-based architecture with a PPO objective, WRT optimizes the non-differentiable NC objective while achieving shape control. Theoretical contributions include new Cheeger inequalities linking spectral and algebraic properties of graph partitions, with good empirical performances against baseline methods.

Reviewers agreed that while interesting, the paper describes an ad-hoc solution to the NC problem with strong assumptions on the nature of the graph, and limited experimental analysis with respect to ablations, size of graphs or competing methods. In this light, I am recommending a reject decision, and I encourage the authors to further strengthen their work on the questions raised by reviewers.

**Additional Comments On Reviewer Discussion:**

Authors implemented significant changes in their paper after the start of the rebuttal process. While I believe that those changes are improving  the quality of the paper, this new version might require another round of review.

---

### Decision · Program_Chairs · 2025-01-22

Reject